# Astrocytic GLUT1 deletion in adult mice enhances glucose metabolism and resilience to stroke

Laetitia Thieren[1,2,9], Henri S. Zanker [1,2,9], Jeanne Droux[2,3], Urvashi Dalvi[1,2], Matthias T. Wyss [1,2], Rebecca Waag [2,4], Pierre-Luc Germain [2,4,5], Lukas M. von Ziegler[2,4], Zoe J. Looser[1,2], Ladina Hösli[1,2], Luca Ravotto [1,2], E. Dale Abel [6], Johannes Bohacek [2,4], Susanne Wegener [2,3], L. Felipe Barros[7,8], Mohamad El Amki [2,3], Bruno Weber [1,2] ✉ & Aiman S. Saab [1,2] ✉

Brain activity relies on a steady supply of blood glucose. Astrocytes express glucose transporter 1 (GLUT1), considered their primary route for glucose uptake to sustain metabolic and antioxidant support for neurons. While GLUT1 deficiency causes severe developmental impairments, its role in adult astrocytes remains unclear. Here, we show that astrocytes and neurons tolerate the inducible, astrocyte-specific deletion of GLUT1 in adulthood. Sensorimotor and memory functions remain intact in male GLUT1 cKO mice, indicating that GLUT1 loss does not impair behavior. Despite GLUT1 loss, two-photon glucose sensor imaging reveals that astrocytes maintain normal resting glucose levels but exhibit a more than two-fold increase in glucose consumption, indicating enhanced metabolic activity. Notably, male GLUT1 cKO mice display reduced infarct volumes following stroke, suggesting a neuroprotective effect of increased astrocytic glucose metabolism. Our findings reveal metabolic adaptability in astrocytes, ensuring glucose uptake and neuronal support despite the absence of their primary transporter.

Astrocytes play an essential role in brain energy metabolism, underscored by their numerous homeostatic functions and metabolic cooperation with neurons and other glial cells[1–3]. Astrocytic endfeet contact blood vessels[4], facilitating glucose uptake through glucose transporter 1 (GLUT1), which is expressed on both endothelial cells and astrocytes[5–7]. In fact, GLUT1 is considered the primary glucose transporter of astrocytes[7–9]. Astrocytic glucose dynamics were shown to be modulated by neuronal activity, implicating GLUT1-mediated glucose uptake by astrocytes to be critical for brain function[10–13]. Glycolytic astrocytes partly metabolize glucose to lactate, which when released serves as signaling molecule and energy substrate for neighboring neurons[2,14,15]. An immediate release of lactate from cortical astrocytes was recently shown to be triggered upon arousal-induced cortical activity in vivo[16]. This process is involved in supporting neuronal functions and is critical for memory consolidation and brain plasticity[14,17].

In Drosophila, glycolytic glial cells were shown to provide neurons with energy substrates fueling their survival[18]. In mice, astrocytes

[1]University of Zurich, Institute of Pharmacology and Toxicology, Zurich, Switzerland. [2]Neuroscience Center Zurich, University and ETH Zurich, Zurich, Switzerland. [3]Department of Neurology, University Hospital and University of Zurich, Zurich, Switzerland. [4]Lab of Molecular and Behavioral Neuroscience, Institute for Neuroscience, Department of Health Sciences and Technology, ETH Zurich, Zurich, Switzerland. [5]Lab of Statistical Bioinformatics, University of Zurich, Zurich, Switzerland. [6]Department of Medicine, David Geffen School of Medicine at UCLA, Los Angeles, CA, USA. [7]Centro de Estudios Científicos (CECs), Valdivia, Chile. [8]Facultad de Medicina, Universidad San Sebastián, Valdivia, Chile. [9]These authors contributed equally: Laetitia Thieren, Henri S. Zanker. ✉e-mail: bweber@pharma.uzh.ch; asaab@pharma.uzh.ch

remain viable despite the disruption of mitochondrial respiration (using conditional Cox10 mutants), suggesting that aerobic glycolysis alone in astrocytes is sufficient to maintain cellular integrity and neural circuit functions[19]. In contrast, deleting Cox10 in neurons leads to neurodegeneration, indicating that oxidative phosphorylation is essential for neuronal survival and glycolysis alone cannot meet the ATP demands[20,21]. Hence, astrocytes appear to be metabolically more flexible than neurons, or glucose metabolism may play a more critical role in astrocyte survival and function. However, the metabolic plasticity of astrocytes remains poorly explored[22], with astrocytic adaptations primarily reported in response to ketogenic diet[23] and in the context of disease and neuroinflammation[24,25].

Glucose hypometabolism has been associated with cognitive decline in neurodegenerative disorders such as Alzheimer's and Parkinson's diseases[26–28]. In a mouse model of Alzheimer's disease, the reduction of endothelial GLUT1 levels and subsequent decrease in brain glucose uptake have exacerbated disease progression and cognitive function[6]. Moreover, GLUT1 deficiency syndrome is a genetic disorder caused by autosomal dominant mutations in the *SLC2A1* gene, leading to a dysfunctional GLUT1 protein that reduces brain glucose availability. This condition is associated with developmental delays, acquired microcephaly, and infantile seizures[29,30]. Homozygous inactivation of the GLUT1 gene (*Slc2a1*) in mice results in embryonic lethality[31]. Inducible deletion of GLUT1 from endothelial cells in adult mice leads to severe neuroinflammation, neuronal loss, and rapid lethality[32,33], underscoring the indispensable role of GLUT1 as the primary endothelial transporter for glucose entry into the brain. However, the specific contributions of astrocytic GLUT1 in maintaining cellular functions and overall brain health have yet to be fully elucidated.

Here, we investigate astrocytic GLUT1 using inducible, astrocyte-specific GLUT1 deletion in adult mice (GLUT1 cKO). Astrocytes lacking GLUT1 remain viable, and GLUT1 cKO mice do not develop any visible signs of glial pathology, neuronal loss, or inflammation. GLUT1 cKO animals show no overt behavioral deficits, as their sensorimotor and cognitive performances are comparable to those of littermate controls. Unexpectedly, glucose uptake and consumption are strongly upregulated in GLUT1-deficient astrocytes. Upon exposure to stroke, GLUT1 cKO mice exhibit a smaller infarct size, suggesting that enhanced glucose metabolism in astrocytes increases resilience to stroke. Taken together, our results demonstrate that astrocytes are capable of adapting their glucose uptake machinery in the absence of GLUT1. We propose that this metabolic plasticity of astrocytes not only maintains cellular and neural circuit functions but also confers neuroprotection.

## Results

### Inducible deletion of astrocytic GLUT1 in adult mice

Astrocytes are considered to rely on GLUT1, their key glucose transporter[8,9,13], to ensure glucose uptake and glycolysis. To investigate the role of GLUT1 in astrocytic glucose metabolism and brain function in adult mice, we generated inducible and astrocyte-specific GLUT1 conditional knockout mice (GLUT1 cKO mice). We crossbred mice carrying loxP-flanked *Slc2a1* (GLUT1[fl/fl])[34] with GLAST[CreERT2/+] mice[35], which express the tamoxifen-inducible Cre-recombinase under the astrocyte-specific GLAST (*Slc1a3*) promoter (Fig. 1a). We induced GLUT1 recombination in 8–10-week-old *GLUT1[fl/fl];GLAST[CreERT2/+]* mice via tamoxifen treatment and conducted experiments approximately 60 days post-injection (Fig. 1b). Littermate control mice (*GLUT1[fl/fl];GLAST[+/+]*) were treated identically with tamoxifen.

Given that endothelial cells express high levels of GLUT1[6], we assessed GLUT1 deletion in GLUT1 cKO mice using western blot analysis on capillary-depleted brain homogenates (Fig. 1c, Supplementary Fig. 1a). As previously reported[6], the 55 kDa isoform of GLUT1 was abundant in brain microvessels, whereas the astrocytic 45 kDa isoform

of GLUT1 was enriched in capillary-depleted brain homogenates (Supplementary Fig. 1b). The latter isoform was significantly reduced by 50% in GLUT1 cKO forebrain lysates relative to controls (Fig. 1c, d), a substantial reduction given that astrocytes account for approximately 20% of the cells in the forebrain[36], and that GLUT1 is also expressed by other neural cell types[8,37,38]. Notably, immunoblot analysis revealed no changes in the abundance of GLUT2, GLUT3, and GLUT4 (Supplementary Fig. 1c–h), suggesting that the deletion of astrocytic GLUT1 did not affect the abundance of these other main glucose transporters. The specific loss of GLUT1 from astrocytes in cKO mice was confirmed by immunohistochemistry (Fig. 1e, f). To assess the extent of astrocytic GLUT1 deletion, we devised an approach to enrich actively translated mRNA specifically from astrocytes by intravenously injecting a Cre-dependent adeno-associated virus (AAV) expressing EGFP-tagged ribosomal protein L10a (EGFPL10a) under the hGFAP promoter (AAV-PHP.eB-hGFAP-DIO-EGFPL10a) into *GLUT1[fl/fl];GLAST[CreERT2/+]* mice and *GLUT1[+/+];GLAST[CreERT2/+]* control mice, followed by tamoxifen treatment 7 days post-AAV injection (Fig. 1g). Immunohistochemistry confirmed widespread cortical expression of EGFPL10a specifically in astrocytes with this approach (Fig. 1h). We examined the enrichment of astrocytic translating mRNAs by Gfap qPCR analysis on samples from cortical lysates, comparing input and immunoprecipitated (IP) samples, and revealed a comparable eight-fold increase in *Gfap* mRNA abundance in IP samples from both genotypes (Fig. 1i). RNA-seq analysis confirmed enrichment of astrocyte-specific translating mRNAs in the IP samples (Supplementary Fig. 2a). Analysis of GLUT1 (*Slc2a1*) mRNA expression in IP samples by qPCR and RNA-seq analysis revealed a 70% reduction in *Slc2a1* transcript abundance in cKO mice compared to controls (Fig. 1j, k). Next, we investigated whether the reduction of GLUT1 could be compensated by other glucose transporters (GLUTs) in astrocytes, indicated by changes in actively translated mRNAs. RNA-seq analysis on IP samples showed no significant differences in mRNA levels of other GLUTs between the genotypes (Fig. 1l). Adult brain tissue can be fueled to some extent by circulating lactate and ketone bodies, which enter the brain via monocarboxylate transporter 1 (MCT1). However, we found no overt changes in the expression levels of *Slc16a1* (MCT1) or the other major MCTs expressed in mammalian cells (Fig. 1l). Differential gene expression analysis identified 13 DEGs that were significantly (FDR < 0.05) up- or downregulated between the genotypes (Supplementary Fig. 2b), reflecting overall moderate changes in the translational profile. Gene Ontology (GO) enrichment analysis highlighted biological processes such as vacuolar acidification, positive regulation of glial cell differentiation, and RNA processing (Supplementary Fig. 2c). Notably, no overt changes were observed in processes related to energy metabolism, suggesting that GLUT1 deletion does not strongly impact the translation of metabolic genes in astrocytes.

### Astrocytic GLUT1 deletion does not lead to behavioral changes

Considering GLUT1's crucial role as the primary glucose transporter in astrocytes, we next investigated whether its loss would adversely affect astrocytic functions, impacting neural viability and integrity. Analysis of RNA-seq data from cortical IP samples, assessing the expression of astrocyte-reactive markers[39], showed no differences in marker expression between genotypes (Supplementary Fig. 3a), indicating absence of astrogliosis following GLUT1 deletion. Similarly, immunoblotting and immunostaining for GFAP, which is typically upregulated in astrogliosis and neuroinflammatory conditions[39,40], revealed no differences in the cKO cortex compared to control animals (Supplementary Fig. 3b–d). Sholl analysis revealed comparable astrocyte morphology between the genotypes (Supplementary Fig. 3e). Astrocytic density remained normal (Supplementary Fig. 3f), and there were no signs of microglial activation, as shown by comparable immunostainings for IBA1, a marker for microglia (Supplementary Fig. 3g). Furthermore, neuronal cell density was similar across

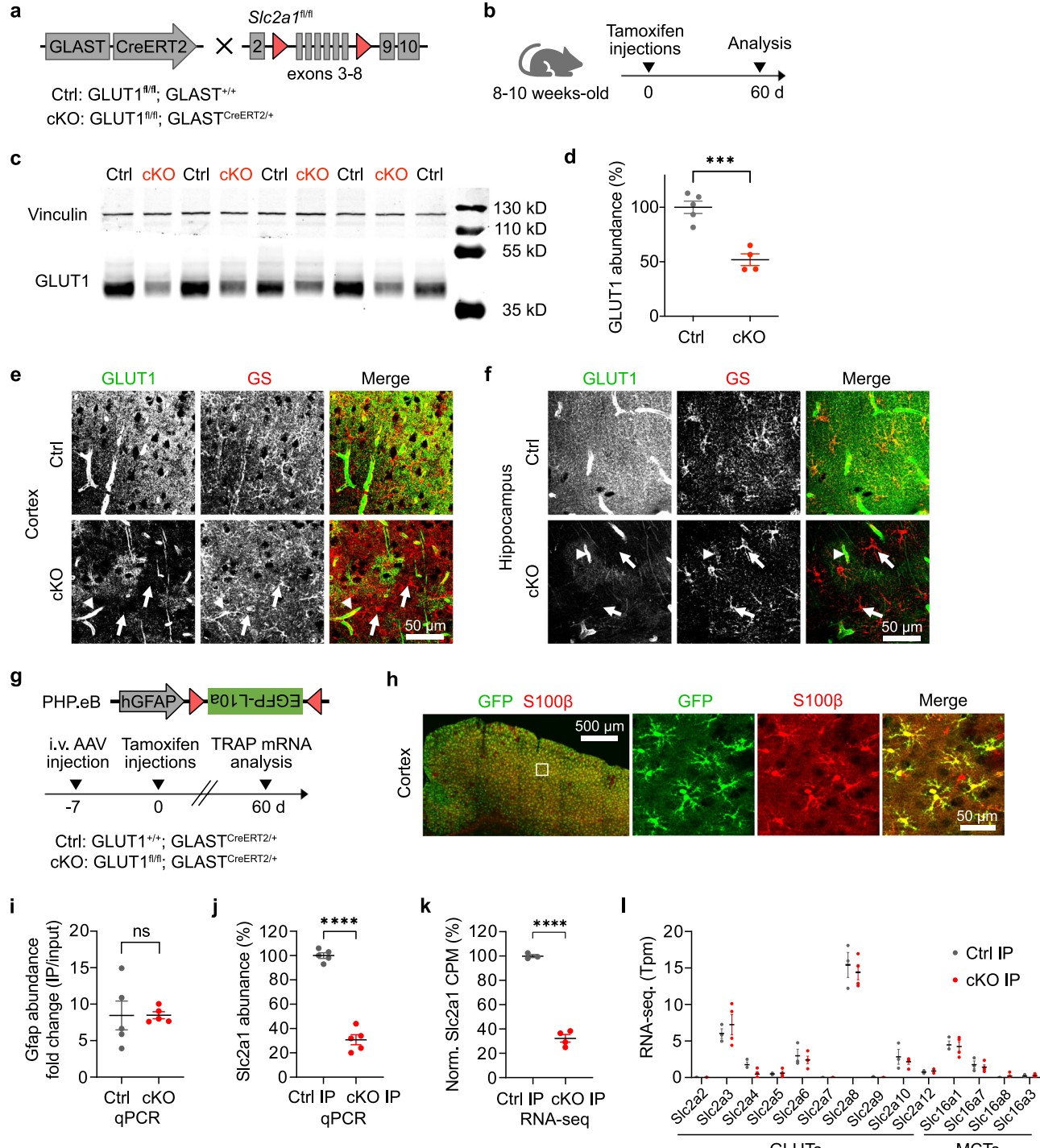

**Fig. 1 | Inducible deletion of astrocytic GLUT1 in adult mice. a** Generation of
GLUT1$^{fl/fl}$;GLAST$^{CreERT2/+}$ (GLUT1 cKO) mice and littermate controls (GLUT1$^{fl/}$
$^{fl}$;GLAST$^{+/+}$). **b** Tamoxifen injections over 5 consecutive days in 8- to 10-week-old
mice, with analyses conducted 60 days post-injection. **c, d** Western blot of GLUT1
(45 kDa) in capillary-depleted brain tissue shows a 48 ± 5% reduction in cKO mice
($n = 4$) compared to controls ($n = 5$, $p = 0.0005$, two-sided unpaired t-test). Vinculin
served as loading control. Immunolabeling for GLUT1 and glutamine synthetase
(GS) in cortex (**e**) and hippocampus (**f**) shows a marked reduction of GLUT1 signal in
cKO brain sections (bottom panels). White arrows highlight GLUT1-depleted
astrocytes, while capillaries (arrowheads) retain GLUT1 expression. Observed in
four mice per genotype. **g** Cre-dependent AAV (PHP.eB-hGFAP-DIO-EGFPL10a)
injected intravenously into *GLUT1$^{fl/fl}$;GLAST$^{CreERT2}$* mice (cKO) and *GLUT1$^{+/}$
$^+$;GLAST$^{CreERT2}$* mice (as controls) enabled astrocyte-specific EGFP-L10 expression
for translating ribosome affinity purification (TRAP). Tamoxifen was administered

7 days post-AAV injection, and TRAP RNA was collected 60 days later.
**h** Immunostaining confirms widespread EGFP-L10 expression (anti-GFP) in astro-
cytes (anti-S100β) using the approach in (**g**), observed in two mice per genotype.
**i** qPCR analysis of astrocytic Gfap polysome-associated RNA following TRAP shows
an 8.5-fold enrichment in immunoprecipitated (IP) samples vs. input in control
($n = 5$) and cKO mice ($n = 5$, $p = 0.9873$, two-sided unpaired t-test). **j** qPCR analysis of
GLUT1 (Slc2a1) polysome-associated RNA in IP samples shows a 69 ± 5% reduction
in cKO mice ($n = 5$) vs. controls ($n = 5$, $p < 0.0001$, two-sided unpaired t-test). **k** RNA-
seq reads (counts per million, CPM) from exons 3–8 of *Slc2a1* show a 68 ± 4%
reduction in cKO cortical IP samples ($n = 4$) vs. controls ($n = 3$, $p < 0.0001$, two-sided
unpaired t test). **l** RNA-seq analysis of glucose and monocarboxylate transporters
reveals no significant differences between genotypes ($n = 3$ vs. 4, $p > 0.85$ for all
genes, two-way Anova with Šídák's multiple comparisons test). Data are presented
as scatter dot plots with mean ± SEM. Source data are provided as a Source Data file.

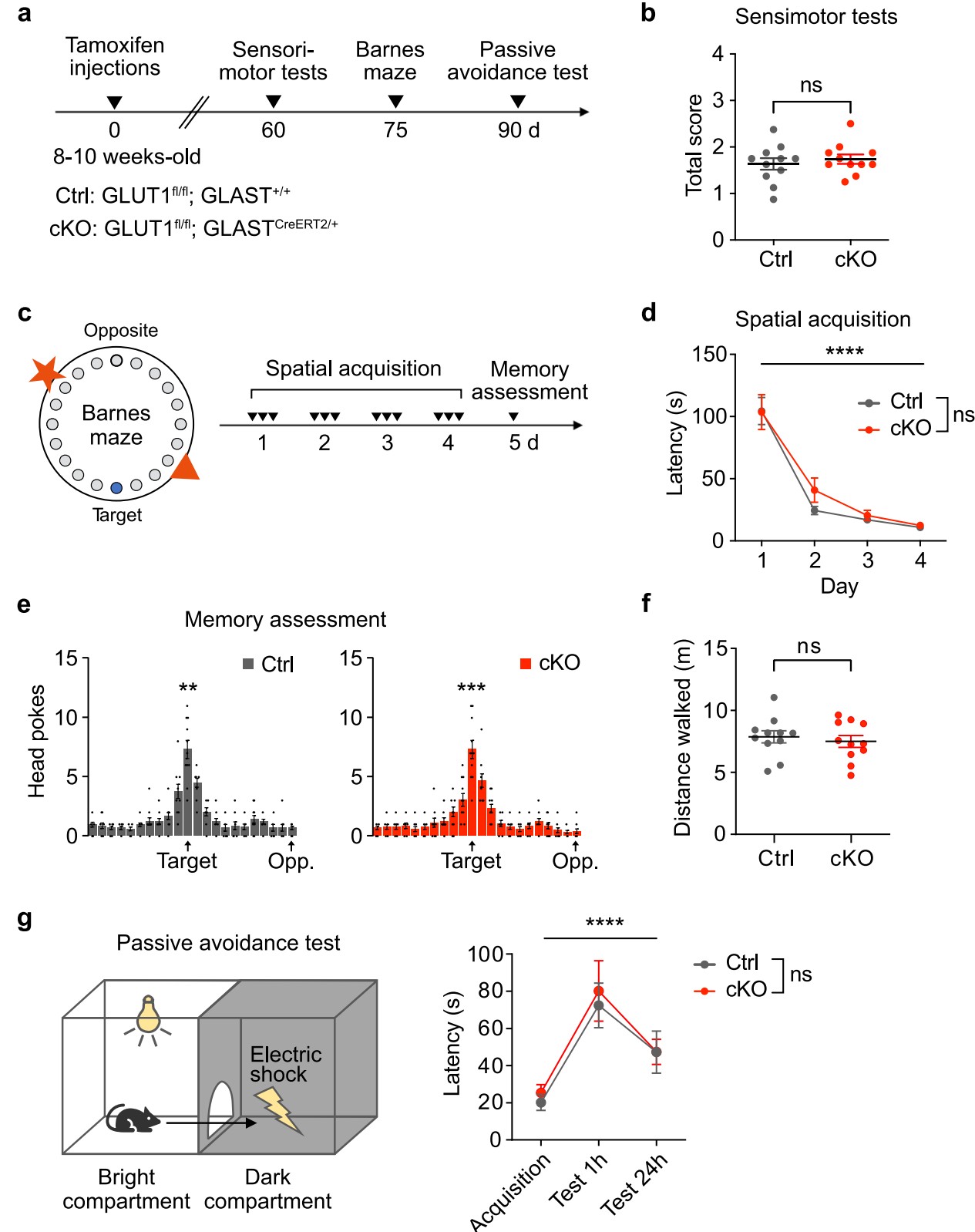

genotypes (Supplementary Fig. 3h). Hence, the loss of astrocytic GLUT1 did not lead to secondary neuroinflammation and degeneration.

To determine whether GLUT1 deletion affected systemic metabolic homeostasis, we assessed body weight and non-fasted blood glucose concentrations, both of which were comparable between male cKO and control mice (Supplementary Fig. 3i, j). This indicates that

astrocytic GLUT1 deletion does not alter basal blood glucose homeostasis.

Since astrocyte dysfunctions are known to impact sensorimotor and cognitive abilities[40], we next examined whether the loss of astrocytic GLUT1 affects behavioral functions. We evaluated sensorimotor skills and cognitive performance in male cKO mice and littermate controls 2-3 months after tamoxifen treatment (Fig. 2a). Sensorimotor

**Fig. 2 | Astrocytic GLUT1 deletion does not affect sensorimotor, learning or memory functions. a** Experimental timeline: Behavioral tests were conducted 60, 75, and 90 days post-tamoxifen treatment, with a one-week rest between tests. **b** Sensorimotor performance (gait, ledge, hindlimb clasping, and horizontal wire test) was comparable between Ctrl ($n = 11$) and cKO ($n = 11$) mice ($p = 0.5295$, two-sided unpaired t-test). **c** Left: Barnes maze setup with 22 holes, one escape hole, and visual cues. Right: Experimental timeline with a 4-day acquisition phase (arrowheads indicate sessions), followed by a day 5 memory test with the target hole sealed. **d** Both Ctrl ($n = 11$) and cKO ($n = 11$) mice showed a significant decrease in latency to find the target hole from day 1 to day 4 (Ctrl: β = -93.71 s, cKO: β = −91.05 s, each $p < 0.0001$, two-sided paired t-test), with no genotype differences in spatial learning ($F_{interaction}(3,60) = 0.6990$, $p = 0.5563$, two-way ANOVA). **e** On day

5, Ctrl ($n = 11$) and cKO ($n = 11$) mice preferentially identified the target hole over an adjacent hole (Ctrl: β = 3.6, $p = 0.0027$; cKO: β = 4.3, $p = 0.0003$, two-sided paired t-test), indicating intact memory retrieval in cKO mice. **f** Locomotor performance on day 5 was comparable between Ctrl ($7.9 \pm 0.5$ m; $n = 11$) and cKO ($7.5 \pm 0.5$ m; $n = 11$, $p = 0.5979$, two-sided unpaired t-test). **g** Passive avoidance setup with bright and dark compartments. Memory retention was measured by latency to enter the dark compartment 1 h and 24 h post-acquisition. Both Ctrl ($n = 11$) and cKO ($n = 10$) mice showed increased latency at 1 h (Ctrl: β = 52.27 s, $p = 0.0015$; cKO: β = 54.90 s, $p = 0.0126$) and 24 h (Ctrl: β = 27.09 s, $p = 0.058$; cKO: β = 22.10 s, $p = 0.0040$), with no genotype differences ($F_{interaction}(2,38) = 0.08586$, $p = 0.9179$, two-way ANOVA). Data are represented as mean ± SEM. Source data are provided as a Source Data file.

performance was assessed through gait analysis, the ledge test, hindlimb clasping, and the horizontal wire test (see Methods for more details), all of which resulted in a comparable total score between control and cKO mice (Fig. 2b). To evaluate spatial learning and memory, we employed the Barnes maze task, wherein mice must learn and remember the location of an escape hole using visual cues (Fig. 2c). The learning curves during the acquisition phase were similar between genotypes, indicating intact spatial learning abilities in cKO mice (Fig. 2d). Furthermore, during memory recall, both control and cKO mice located the target hole with equal proficiency, confirming intact memory formation and retention (Fig. 2e). Additionally, both control and cKO mice walked comparable distances, which confirms normal locomotor activity (Fig. 2f).

We also conducted the passive avoidance test (Fig. 2g), a fear-motivated learning paradigm, given the reported importance of astrocyte metabolism in fear-related memory formation[17]. During the acquisition phase, normal latencies were observed between genotypes in crossing into the dark compartment, a natural behavior in mice, upon which they received an electric foot shock delivered to the grid floor. When tested 1 and 24 h later, both control and cKO mice demonstrated significantly longer latencies to cross into the dark compartment, indicating no difference in short- and long-term memory formation between the genotypes (Fig. 2g).

The absence of detectable impairments in learning and memory in GLUT1 cKO animals suggests that astrocytic GLUT1 deletion does not overtly disrupt cognitive function under these conditions.

### Glucose metabolism is enhanced in GLUT1 cKO astrocytes

Given the absence of neuroinflammation and behavioral deficits in GLUT1 cKO mice, we explored whether glucose uptake and metabolism are affected in astrocytes lacking GLUT1. To address this, we utilized the genetically-encoded glucose FRET sensor FLII12 Pglu700μΔ6 (termed FLIIP)[41] and first assessed its functionality in cortical slices of adult wild-type mice following intracortical delivery via AAVs encoding FLIIP under the hGFAP promoter (Supplementary Fig. 4a). Immunohistochemistry for S100β confirmed astrocyte-specific expression of FLIIP 3 weeks after AAV delivery (Supplementary Fig. 4b). Two-photon (2P) sensor imaging in acute brain slices was performed 3–4 weeks after intracortical AAV injections in artificial cerebrospinal fluid (ACSF) containing 5 mM glucose and 3 mM lactate as energy substrates. Notably, astrocytic glucose levels significantly increased upon elevating extracellular glucose concentrations ([Glc]$_O$) to 50 mM and decreased upon removal of glucose from the ACSF (Supplementary Fig. 4c, d). Hence, the sensor in astrocytes is not saturated at 5 mM [Glc]$_O$ and allows the study of astrocytic glucose metabolism.

Next, we investigated how glucose metabolism was affected upon GLUT1 deletion. To ensure we primarily imaged astrocytes targeted for Cre recombination in GLUT1 cKO mice, we co-injected a Cre-reporter AAV encoding TdTomato (AAV2/8-CAG-DIO-TdTomato) together with an AAV encoding FLIIP (AAV2/9-hGFAP-FLIIP). One week later, we

started the tamoxifen treatment for 5 consecutive days and performed acute brain slice experiments 60-70 dpi (Fig. 3a). Control mice also expressed CreERT2 (*GLUT1$^{+/+}$; GLAST$^{CreERT2/+}$*), and only astrocytes expressing both TdTomato and FLIIP in both genotypes (Fig. 3b) were used for analysis. Notably, we verified that the Cre-reporter AAV was not leaky and only expressed TdTomato in Cre-active astrocytes, given the absence of TdTomato in *GLAST$^{+/+}$* mice lacking CreERT2 expression (Supplementary Fig. 5). To assess basal glucose levels, we obtained the raw FRET ratios of individual cortical astrocytes from control and cKO brain slices at 5 mM [Glc]$_O$ (Fig. 3c, d). For subsequent normalization to the minimum, glucose levels were also assessed at 0 mM [Glc]$_O$ (Fig. 3c, d). Both the raw and normalized FRET ratios (Fig. 3d, e) showed no difference between the genotypes, suggesting that GLUT1 cKO astrocytes maintained comparable resting glucose levels to control astrocytes.

Resting glucose levels are controlled by the balance between glucose uptake and glucose consumption rates. Given this, we investigated whether GLUT1-deficient astrocytes would exhibit altered glucose uptake and metabolism. To assess glucose uptake dynamics, we transiently increased [Glc]$_O$ from 5 to 25 mM, which increased cytosolic glucose levels in both GLUT1 cKO and control astrocytes (Fig. 4a). The slope of glucose increase was similar between the genotypes (Fig. 4b), indicating comparable glucose uptake kinetics. Interestingly, the increase in intracellular glucose levels was significantly higher in cKO compared to control astrocytes (Fig. 4c). This finding is counterintuitive, as GLUT1 is considered the primary glucose transporter in astrocytes.

Next, to determine if glycolytic activity was altered in GLUT1 cKO mice, we used a transport-stop assay employing cytochalasin B (CytoB), a glucose transporter blocker, to measure the intracellular glucose consumption rate[42] (Fig. 4d–f). Unexpectedly, GLUT1 cKO astrocytes exhibited a 2.6-fold increase in glycolytic rate compared to controls (Fig. 4e, f). This indicates that GLUT1 cKO astrocytes have shifted towards higher glucose metabolism compared to control astrocytes. Notably, higher glucose consumption in the face of preserved steady-state glucose (Fig. 3) points to higher glucose permeability. Together, these findings suggest a substantial metabolic adaptation in GLUT1-deficient astrocytes, upregulating both glucose uptake and consumption.

Given that we did not detect overt upregulation of other glucose transporters in GLUT1 cKO mice (Fig. 1l, Supplementary Fig. 1), we explored whether astrocytes lacking GLUT1 might utilize alternative glucose uptake mechanisms. One possibility is that GLUT1-deficient astrocytes could receive glucose from neighboring, non-recombined (~30%) astrocytes (still expressing GLUT1) through the gap-junction coupled network[43,44]. Additionally, although not previously reported in astrocytes, connexin hemichannel-mediated glucose uptake has been proposed as an alternative pathway in cultured oligodendrocyte precursor cells[45], which may potentially become 'activated' in GLUT1-deficient astrocytes. To investigate these possibilities, we examined astrocytic glucose dynamics in the presence of carbenoxolone (CBX), a

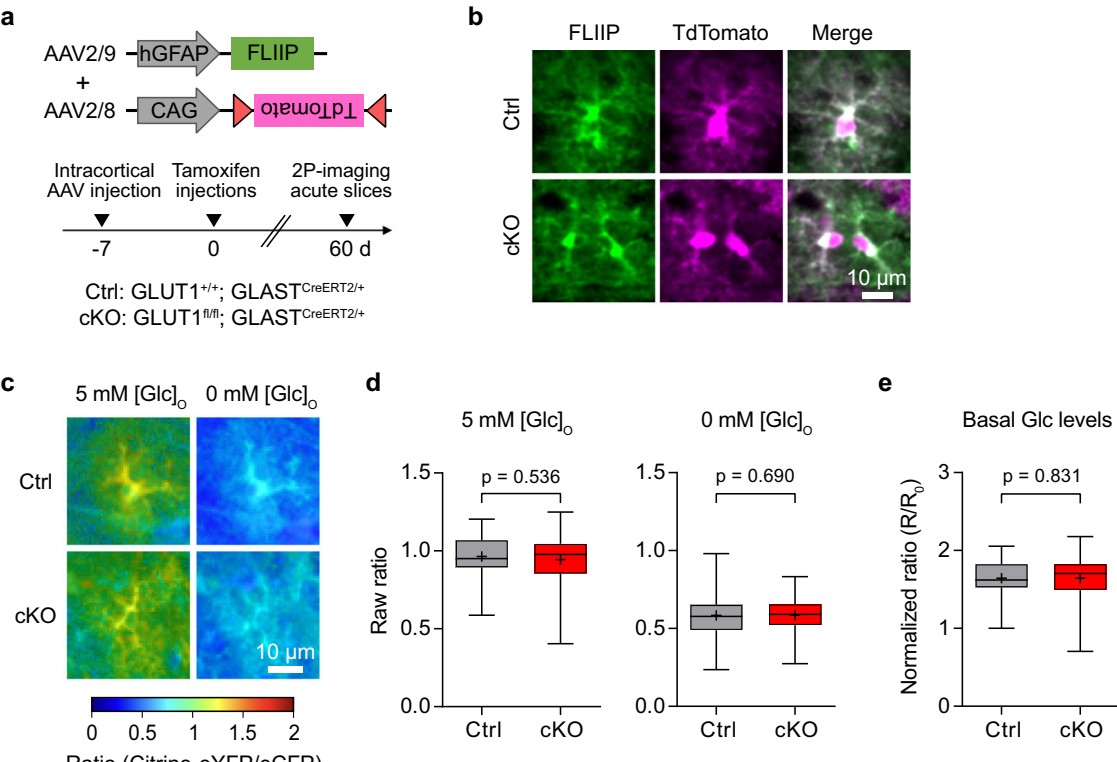

**Fig. 3 | Astrocytes have normal basal glucose levels in GLUT1 cKO mice.**
**a** Experimental timeline: Intracortical injection of AAVs encoding glucose sensor FLII12 Pglu700μΔ6 (FLIIP) and DIO-TdTomato into ~8-week-old *GLUT1^fl/fl^;GLAST^CreERT2/+* (cKO) and *GLUT1^+/+^;GLAST^CreERT2/+^* (Ctrl) mice, followed by tamoxifen treatment 7 days later. Acute cortical slice two-photon (2 P) imaging was performed ~60 days post-tamoxifen treatment. **b** Example 2P images of cortical astrocytes co-expressing FLIIP and TdTomato (marking cells targeted for Cre recombination) selected for glucose imaging (observed in 5 Ctrl and 6 cKO mice). Scale bar: 10 μm. **c** Representative color-coded glucose sensor ratio images from Ctrl and cKO astrocytes in ACSF with 5 mM glucose ([Glc]$_O$) and after removal of extracellular glucose (0 mM [Glc]$_O$). Warm and cold colors indicate high and low glucose levels, respectively. **d** Quantification of raw sensor ratios from astrocytes under conditions in (**c**). At 5 mM [Glc]$_O$ and 0 mM [Glc]$_O$, glucose sensor ratios were comparable between Ctrl ($n = 63$ cells from 9 slices, 5 mice) and cKO ($n = 101$ cells from 12 slices, 6 mice) ($p = 0.536$ at 5 mM [Glc]$_O$; $p = 0.690$ at 0 mM [Glc]$_O$, two-sided linear mixed model). **e** Normalized glucose sensor ratios (5 mM [Glc]$_O$ ratios normalized to the average minimum at 0 mM [Glc]$_O$) show comparable basal glucose levels between genotypes ($p = 0.831$, two-sided linear mixed model). Box plots show the median (center line), quartiles (box bounds), mean (+) and min-to-max (whiskers). Source data are provided as a Source Data file.

blocker of both hemichannel activity and gap junction coupling[43]. Interestingly, bath application of CBX significantly reduced cytosolic glucose levels (Supplementary Fig. 6a), with a decrease occurring at a similar rate in both genotypes (Supplementary Fig. 6b). We then assessed if glucose uptake differed between the genotypes in the presence of CBX by transiently increasing [Glc]$_O$ from 5 mM to 25 mM. No difference in glucose uptake kinetics was observed between the genotypes (Supplementary Fig. 6c). Yet, even with CBX, increase in intracellular glucose levels remained significantly higher in cKO compared to control astrocytes (Supplementary Fig. 6d). These findings indicate that the enhanced glucose uptake in GLUT1 cKO astrocytes is not compensated for by the gap junction-coupled network or increased connexin hemichannel activity. The reduction in glucose levels following CBX application suggests that gap junction coupling and/or connexin hemichannel functions play a role in maintaining glucose homeostasis in astrocytes, a process that appears to be independent of GLUT1.

## Stroke-induced neural injury is reduced in GLUT1 cKO mice

An enhanced glucose metabolism in astrocytes could be neuroprotective in conditions of stroke, possibly by facilitating increased lactate supply to neurons, which serves as an alternative energy source, and by augmenting the production of glutathione, crucial for mitigating oxidative stress[46,47]. To test this hypothesis, we subjected male GLUT1 cKO and control mice to thrombin-induced occlusion of the middle cerebral artery (MCA) (Fig. 5a), employing a reliable stroke model in mice[48]. Laser speckle imaging (LSI) conducted before and shortly after stroke induction confirmed that the occlusion-induced hypoperfusion was similar in both groups (Fig. 5b, c), ensuring a valid comparison of stroke-induced lesions between the genotypes. Post-stroke body weight changes, indicative of overall health and stress levels, were comparable between control and GLUT1 cKO mice (Fig. 5c). One week after stroke, animals were perfused for histological analysis of neural injury volumes (Fig. 5e). GLUT1 cKO mice exhibited significantly smaller lesion volumes (~43% reduction) compared to littermate controls (Fig. 5f), indicating that the elevated astrocytic glucose metabolism in GLUT1 cKO associates with a neuroprotective effect. Both genotypes showed reactive gliosis, with infarct border astrocytes extending processes toward the infarct core, surrounded by dense, rounded macrophages and microglia (Fig. 5g). The infarct border (0–150 μm from the infarct edge) was characterized by astrocytes with elongated, core-oriented processes, while the peri-infarct region (150–500 μm beyond the border) contained reactive astrocytes with a less elongated morphology. Sholl analysis confirmed similar elongated astrocytic morphology at the infarct border in both genotypes (Fig. 5h), whereas in the peri-infarct region, GLUT1 cKO astrocytes were less ramified than controls (Fig. 5i). In contrast, microglial morphology in the peri-infarct region remained unchanged between genotypes (Supplementary Fig. 7). The less reactive peri-infarct astrocyte profile in GLUT1 cKO mice may indicate a dampened astrocytic response, potentially contributing to the reduced lesion size and neuroprotection observed in these animals.

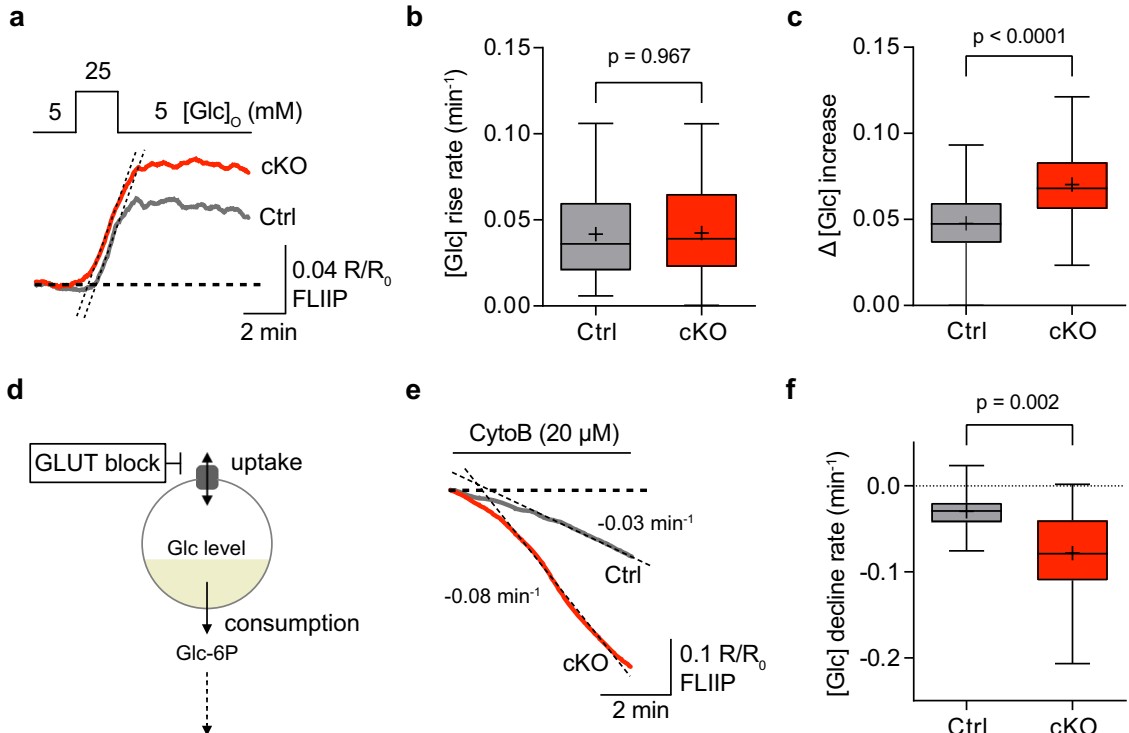

**Fig. 4 | GLUT1 cKO astrocytes have enhanced glucose metabolism.**
**a** Representative glucose sensor traces showing the astrocytic glucose level increase in response to transient elevation of [Glc]$_O$ from 5 mM to 25 mM in Ctrl and cKO brain slices. **b** Glucose rise rate (dashed lines in (**a**)) was similar between Ctrl ($n = 53$ cells from 9 slices, 5 animals) and cKO ($n = 88$ cells from 12 slices, 6 animals, $p = 0.967$, two-sided linear mixed model). **c** Glucose increase ($\Delta$[Glc]) was significantly higher in cKO astrocytes ($n = 88$ cells from 12 slices, 6 animals) compared to Ctrl ($n = 53$ cells from 9 slices, 5 animals, $p < 0.0001$, two-sided linear mixed model). **d** Schematic of glucose consumption assay: Glucose uptake was inhibited

with the GLUT blocker cytochalasin B (CytoB) to assess glucose consumption rate in astrocytes. Glc-6P, glucose-6-phosphate. **e** Representative traces showing the astrocytic glucose decline in Ctrl and cKO brain slices during CytoB incubation. **f** Glucose consumption rate, measured as the decline in [Glc] (dashed lines in (**e**)), was 2.6-fold higher in cKO astrocytes ($-0.079 \pm 0.003$ min$^{-1}$, $n = 101$ cells from 12 slices, 6 animals) compared to Ctrl ($n = 63$ cells from 9 slices, 5 animals, $-0.030 \pm 0.003$ min$^{-1}$, $p = 0.002$, two-sided linear mixed model). Box plots show the median (center line), quartiles (box bounds), mean (+) and min-to-max (whiskers). Source data are provided as a Source Data file.

## Discussion

Astrocyte metabolism is involved in numerous homeostatic functions that ensure neural activity and health[1,2,14,49]. In this study, we demonstrated that the inducible astrocyte-specific deletion of GLUT1 – the key glucose transporter for astrocytes – in adult mice is well-tolerated by both astrocytes and neurons, and does not lead to observable deficits in sensorimotor or cognitive functions. Contrary to expectations, the lack of GLUT1 did not impair astrocytic glucose uptake but was accompanied by a robust increase in both glucose uptake and metabolism. This suggests the presence of an alternative glucose transporter in astrocytes, capable of compensating for glucose uptake and metabolism, that is not necessarily elevated at the translational level when GLUT1 is absent. Moreover, GLUT1 cKO mice showed smaller infarct volumes following stroke, indicating a neuroprotective effect associated with enhanced glucose metabolism in astrocytes. Our study highlights the metabolic resilience of astrocytes in maintaining glucose uptake and metabolism, thereby supporting neural functions.

GLUT1 is considered the master controller of glucose uptake and consumption in the brain[7,50]. This role is primarily associated with endothelial cells, as GLUT1 deletion from endothelial cells results in severe neuroinflammation, neuronal loss, and rapid lethality[32,33]. Interestingly, despite being the most abundantly expressed glucose transporter in astrocytes[8,51], our results demonstrate that GLUT1 is not always the limiting factor for astrocytic glucose uptake. At the translational level, we found that adult astrocytes also express *Slc2a3* (GLUT3) and *Slc2a8* (GLUT8), consistent with earlier findings[51]. However, these transporters are not visibly upregulated when GLUT1 is

deleted from astrocytes. It is possible that their surface expression and/or intrinsic activity is elevated in GLUT1 cKO astrocytes.

In fact, glucose uptake in GLUT1-deficient astrocytes must be mediated by a transport mechanism sensitive to cytochalasin B, which inhibits other glucose transporters, including GLUT3[52,53] and GLUT8[54]. GLUT8 is primarily associated with intracellular membranes, colocalizing with endosomal and lysosomal compartments[55], but appears to undergo insulin-mediated translocation to the plasma membrane[56]. However, GLUT8 localization, surface dynamics, and function in adult astrocytes remains elusive[57,58]. While GLUT3 is primarily considered the main glucose transporter in neurons[53], its specific deletion from neurons using CamK2a-Cre mice did not result in neurodegeneration[59,60], whereas its neural inactivation using Nestin-Cre (causing GLUT3 deletion in glial cells and neurons during embryonic and postnatal development) causes severe developmental perturbations, reduced brain size, and postnatal lethality within 3–4 weeks of age[59]. The role of GLUT3 in astrocytes, however, is still poorly understood. Given that GLUT1 is not the exclusive glucose transporter in adult astrocytes and that glucose uptake is further enhanced when GLUT1 is deleted, future studies should further explore the functional roles of other glucose transporters, including GLUT3 and GLUT8, in astrocytes. These unknown compensatory mechanisms are of clinical interest, for even milder GLUT1 reductions in brain endothelium are not adequately compensated, leading to neurological diseases[6,30].

A key finding of our study is that GLUT1-deficient astrocytes increased their glycolytic activity by 2.6-fold, suggesting that GLUT1 activity or associated mechanisms may regulate glucose metabolism.

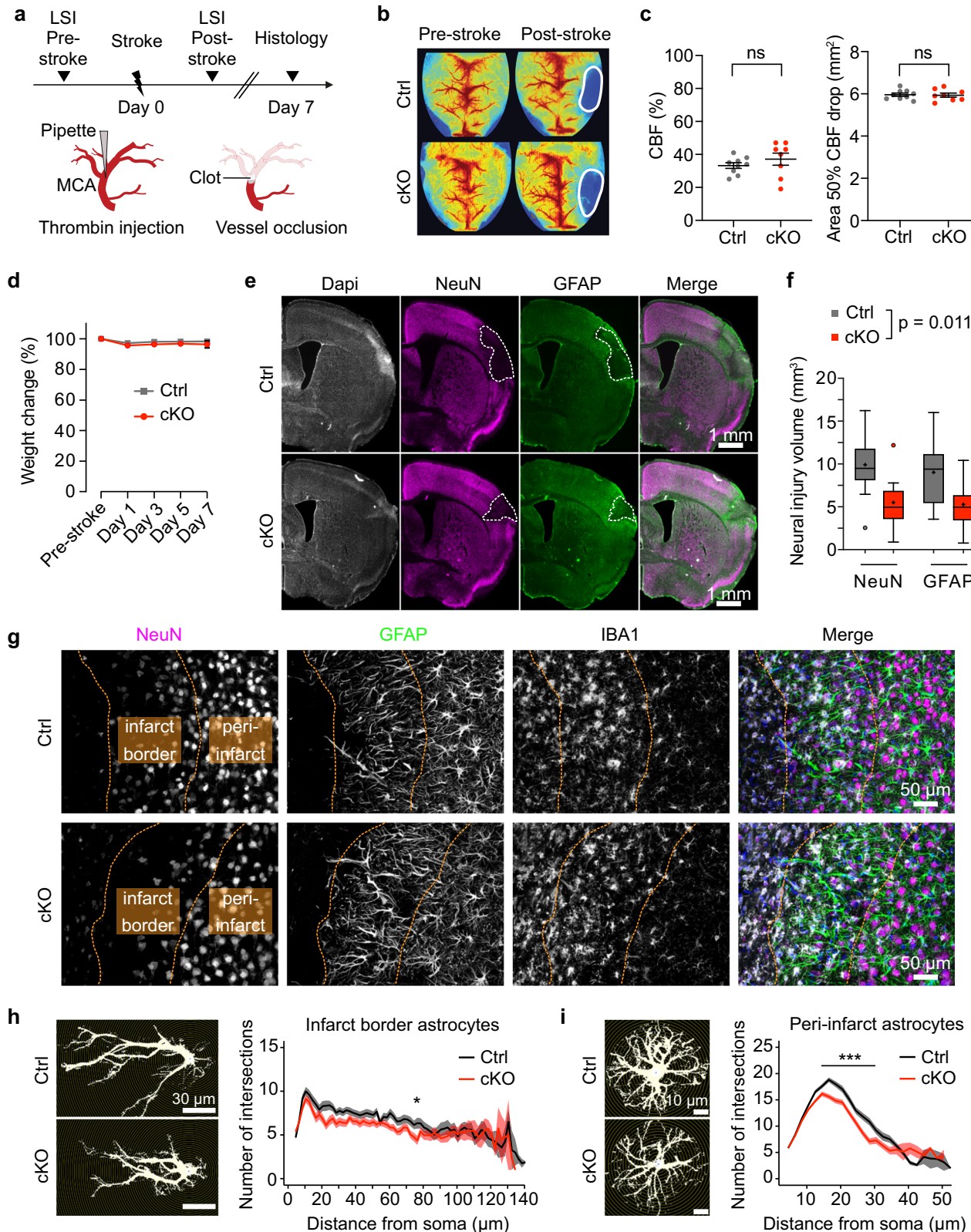

GLUT1 interacts with various proteins and signaling complexes that regulate its function[7]. The absence of GLUT1 might disrupt these interactions, leading to changes in signaling pathways that enhance glucose uptake and consumption. In this sense, GLUT1 may be hypothesized to be a negative modulator of glycolysis, a role that the compensatory mechanism(s) may not be able to play. For instance, the loss of GLUT1 could affect AMP-activated protein kinase (AMPK)

pathway activity or interactions with caveolin-1 or integrins, altering cell signaling and metabolism[61,62]. Deletion of GLUT1 may also result in a perturbation in the balance between glycolysis and oxidative phosphorylation, leading to a shift towards enhanced glycolytic activity. Inhibition of mitochondrial respiration in cultured astrocytes results in elevated glucose metabolism[63], and mice with respiration-deficient astrocytes revealed a 1.4-fold increase in brain lactate concentrations[19],

**Fig. 5 | Reduced stroke-induced neural injury in GLUT1 cKO mice.**
**a** Experimental timeline: Stroke was induced in male mice 60 days post-tamoxifen, followed by perfusion 7 days later. MCA, middle cerebral artery; LSI, laser speckle imaging. **b** Representative (LSI) images of Ctrl and cKO mice pre- and post-stroke, with infarct areas outlined. **c** Cerebral blood flow (CBF) changes (%) (left) and hypoperfused area size (right) showed no significant differences between Ctrl ($n = 9$) and cKO ($n = 8$, $p = 0.3353$ and $p = 0.8724$, two-sided unpaired t-tests). **d** Post-stroke body weight changes did not differ between genotypes ($n = 9$ vs. 8, $F_{interaction}(4,60) = 2.436$, $p = 0.7515$, two-way ANOVA). **e** NeuN (neurons), GFAP (astrocytes) and DAPI (nuclei) stainings of infarcted brain sections 7 days post-stroke. Dashed outlines indicate lesion areas. **f** Neural injury volumes (in mm³) were significantly reduced in cKO ($n = 8$) vs. Ctrl ($n = 9$) when quantified using NeuN and GFAP ($p = 0.011$, linear mixed model). Individually, NeuN-based injury volume was 9.91 mm³ in Ctrl and 5.50 mm³ in cKO, while GFAP-based volumes were 9.02 mm³

in Ctrl and 5.26 mm³ in cKO. On average, cKO mice had 43.1% ± 8.4% SEM smaller lesions. Box plots show the median (center line), quartiles (box bounds), mean (+), and 1.5× interquartile range (whiskers). **g** Immunolabelling of NeuN, GFAP, IBA1 (microglia), and DAPI at the infarct. Dotted lines indicate infarct border and peri-infarct region (observed in 7-8 mice per genotype). **h** Sholl analysis of infarct border astrocytes showed a comparable number of intersections between genotypes, with a significant difference observed at 76 μm from the soma ($n = 48$ vs. 56 cells from 7 and 8 animals, $p = 0.03$; linear mixed-effects model with post-hoc pairwise comparisons). **i** Sholl analysis of peri-infarct astrocytes revealed a significantly fewer intersections in cKO compared to Ctrl ($n = 97$ vs. 110 cells from 7 and 8 animals, $p < 0.001$ within 16–30 μm from the soma; linear mixed model with post-hoc pairwise comparisons). Data in (**c**, **d**, **h**, and **i**) are represented as mean ± SEM. Source data are provided as a Source Data file.

also underscoring the metabolic adaptability of astrocytes to maintain brain functions.

Notably, our findings of enhanced glucose metabolism in astrocytes align well with a recent report that was published during the revision of our study, which demonstrated that astrocytic GLUT1 deletion is associated with increased whole-brain glucose metabolism, as identified by [18]F-FDG PET imaging[64]. However, in contrast to both their PET results and our own findings, their analysis of [U-13C]glucose incorporation in isolated astrocytes suggested a reduction in astrocytic glucose metabolism[64]. This discrepancy likely stems from methodological limitations. Their approach relied on prolonged systemic tracer infusion and a lengthy subsequent astrocyte isolation protocol, during which cellular metabolism remains active. This could lead to label dilution and metabolite turnover, potentially underestimating intracellular glucose metabolism. By contrast, our biosensor imaging provides direct, cell-specific evidence of increased glucose uptake and metabolism in GLUT1-deficient astrocytes. Hence, both our data and their PET findings[64] support the conclusion that astrocytes adaptively enhance their metabolic activity following GLUT1 loss.

Astrocytes form large gap junction-coupled networks that are critical for neural homeostasis and behavior[40]. Astroglial network-mediated glucose shuttling has been proposed under pathological conditions, such as in glaucoma[44]. Interestingly, we observed that astrocytic glucose levels declined upon inhibition of gap junctions and/or hemichannels with CBX in both control and cKO astrocytes at a similar rate. However, this effect is complex, as glucose uptake remained significantly enhanced in cKO astrocytes even in the presence of CBX. This indicates that glucose uptake in GLUT1 cKO astrocytes is not compensated by gap junction-coupled network activity or hemichannel functions. Previous studies have reported that inhibiting gap junction coupling impacts glucose homeostasis in cultured astrocytes, where it was associated with increased glucose uptake and utilization via GLUT1 upregulation to promote astrocyte proliferation[65–67]. However, the role of a connexin-mediated regulation of astrocytic glucose metabolism in adult astrocytes, likely independent of GLUT1, is an important question for future investigation.

We found that the increased glucose metabolism in astrocytes lacking GLUT1 associates with a neuroprotective effect in stroke. This effect may result from various glycolysis-derived compounds released from astrocytes that support and protect neuronal functions[2,14,46]. Enhanced glucose metabolism likely increases lactate production and release. Lactate has been shown to exert neuroprotective effects in ischemic stroke through various mechanisms, including acting as an alternative energy substrate for neurons, modulating neuronal excitability, and reducing oxidative stress[47,68–72]. Additionally, elevated production of L-serine, which has been reported to have neuroprotective benefits, could contribute to the observed effects[73–75]. Moreover, astrocytic glucose can be diverted into the pentose phosphate pathway to produce more NADPH, which is crucial for maintaining the reduced state of glutathione, an antioxidant that protects astrocytes

against oxidative stress-induced dysfunctions[46,76]. Furthermore, astrocytes can supply neurons with precursors which are necessary for the synthesis of glutathione within neurons, enhancing their ability to detoxify reactive oxygen species and maintain redox balance[77,78]. Overall, by enhancing glucose metabolism in astrocytes, these pathways - ranging from metabolic support and neuroprotective signaling to antioxidant protection - likely contribute to the neuroprotective effect observed in GLUT1 cKO mice exposed to ischemic stroke. Taken together, our findings underscore that glucose uptake and metabolism in astrocytes does not solely rely on GLUT1, that astrocytes possess substantial metabolic resilience, and that enhancing astrocytic glucose metabolism confers neuroprotection.

## Methods
### Animals
All experimental procedures were performed according to the guidelines of the Swiss Animal Protection Law, Veterinary Office, Canton Zurich (Animal Welfare Act, 16 December 2005 and Animal Welfare Ordinance, 23 April 2008; licenses ZH169_2017, ZH191_2020). Mice carrying the floxed *Slc2a1* allele [GLUT1[fl/fl] (ref. 34)] were crossbred with GLAST[CreERT2/+] (ref. 35) mice to obtain GLUT1[fl/fl];GLAST[CreERT2/+] mice. Control animals were either GLUT1[fl/fl];GLAST[+/+] or GLUT1[+/+];GLAST[CreERT2/+] mice. Mice were maintained on a C57BL/6 background. For genotyping the following primers were used: for GLUT1 flox; 5′-CTGTGAGTTCCT-GAGACCCTG-3′ and 5′-CCCAGGCAAGGAAGTAGTTC-3′ and for GLAST-CreERT2: 5′- GAGGCACTTGGCTAGGCTCTGAGGA-3′, 5′-GAGGAGATC CTGACCGATCAGTTGG-3′, and 5′-GGTGTACGGTCAGTAAATTGGACAT-3′. For experiments in wild-type animals, Charles River C57BL/6J mice (2-3 months old) were used. Mice were group-housed in standardized cages and kept on an inverted 12 h light/12 h dark cycle at 23 °C and 55% humidity. Food and water were available ad libitum. For behavioral and stroke-related experiments, only male mice were used to minimize variability associated with sex differences in stroke outcomes and behavior. For all other analyses, both sexes were included whenever possible, based on cohort availability.

### Tamoxifen treatment
To induce recombination and GLUT1 deletion, mice received intraperitoneal (i.p.) injections of tamoxifen (100 mg/kg) for five consecutive days. Tamoxifen (T5648, Sigma-Aldrich) was dissolved in corn oil (C8267, Sigma-Aldrich) at 10 mg/ml and freshly prepared for each experimental cohort injected simultaneously. Mice were injected at 8-10 weeks of age, and experiments were conducted from 60 to 90 days post-injection.

### Western blotting
Tissue lysis was performed using 0.5 mm Zirconium Oxide beads (ZROB05, Next Advance) and a Bullet Blender tissue homogenizer (BBX24, Next Advance) in RIPA buffer (150 mM NaCl, 0.1% Triton X-100, 0.5% sodium deoxycholate, 0.1% SDS, 50 mM Tris-HCl, pH 8.0)

**Table 1 | Antibody information**

| Antibody | Host species, type | Method, dilution | Source, Cat. no. |
|---|---|---|---|
| anti-GLUT1 | Rabbit, polyclonal | IHC, 1:300 IB, 1:15,000 | produced by Kathrin Kusch[85] |
| anti-GLUT2 | Rabbit, polyclonal | IB, 1:15,000 | Abcam, Cat. no. ab54460 |
| anti-GLUT3 | Rabbit, monoclonal | IB, 1:15,000 | Abcam, Cat. no. ab191071 |
| anti-GLUT4 | Rabbit, polyclonal | IB, 1:15,000 | Millipore, Cat. no. 07-1404 |
| anti-GS | Mouse, monoclonal | IHC, 1:700 | BD Transduction Laboratories, Cat. no. 610518 |
| anti-GFAP | Chicken, polyclonal | IHC, 1:2,000 | Abcam, Cat. no. Ab4674 |
| anti-GFAP | Rabbit, polyclonal | IHC, 1:1,000 IB, 1:1,000 | DAKO, Cat. no. Z334 |
| anti-IBA1 | Rabbit, polyclonal | IHC, 1:1,000 | FUJIFILM Wako Chemicals, Cat. no. 019-19741 |
| anti-IBA1 | Goat, polyclonal | IHC,1:1,000 | Abcam, Cat. no. ab5076 |
| anti-S100β | Rabbit, monoclonal | IHC, 1:700 | Abcam, Cat. no. ab52642 |
| anti-NeuN | Rabbit, monoclonal | IHC, 1:1,500 | Abcam, Cat. no. ab177487 |
| anti-mouse Alexa 488 | Donkey, polyclonal | IHC, 1:700 | Jackson ImmunoResearch, 715-545-150 |
| anti-rabbit Cy3 | Donkey, polyclonal | IHC, 1:700 | Jackson ImmunoResearch, 711-165-152 |
| anti-chicken Alexa 488 | Donkey, polyclonal | IHC, 1:700 | Jackson ImmunoResearch, 703-545-155 |
| anti-goat Cy5 | Donkey, polyclonal | IHC, 1:700 | Jackson ImmunoResearch, 705-175-147 |
| anti-rabbit IRDye 800CW | Donkey, polyclonal | IB, 1:10,000 | LI-COR, Cat. no.926-32213 |

IHC immunohistochemistry, IB immunoblot.

supplemented with protease inhibitors (11697498001, Roche). Protein concentration was determined using a BCA protein assay (71285-3, Merck Millipore). For Western blotting, 20 or 40 μg of protein lysate was separated by SDS-PAGE on 12% polyacrylamide gels and transferred to a nitrocellulose membrane (0.1 μm pore size, GE10600000, Amersham Protran). The membrane was blocked for 1 h in 5% Blocking Reagent (11921673001, Roche) in PBS, then incubated overnight at 4 °C with primary antibodies (Table 1). The next day, the blot was rinsed with 0.05% PBS-Tween and incubated for 1 h with IRDye 800CW anti-rabbit secondary antibody (1:10,000, LI-COR). Infrared fluorescence was detected using an Odyssey CLx Imaging System (LI-COR). Normalization was performed using Revert 700 Total Protein Stain (926-11016, LI-COR). For GLUT quantification, fresh brain tissue was homogenized in artificial cerebrospinal fluid (ACSF) (in mM: 140 NaCl, 4 KCl, 1 MgCl$_2$, 2 CaCl$_2$, 10 glucose, 10 HEPES, pH 7.4) using a glass tissue grinder and pelleted. Microvessels were depleted from total brain homogenates using dextran density gradient centrifugation as previously described[6]. Briefly, homogenates were mixed with 17% dextran solution (31392, Sigma-Aldrich) in ACSF. After centrifugation, the top layer containing capillary-depleted brain homogenates was collected and processed as described above, while the pellet containing blood vessels was used as a control for endothelial GLUT1 expression.

## Immunohistochemistry

Mice were anesthetized with pentobarbital and transcardially perfused with ice-cold ACSF (10 mL), followed by 2% paraformaldehyde (PFA) (Paraformaldehyde Granular, 19210, Electron Microscopy Sciences) in phosphate-buffered saline (PBS, pH 7.4) (10X Dulbecco's Powder, A0965, Axon Lab AG). After perfusion, brains were post-fixed for 3 h in 4% PFA at 4 °C, then cryoprotected in 30% sucrose in PBS for 2 days at 4 °C. Coronal sections (40 μm) were cut using a microtome (Hyrax KS 34) and stored in antifreeze solution (50 mM sodium phosphate buffer, 1 M glucose, 35% ethylene glycol, 3.5 mM sodium azide, pH 7.4) at −20 °C until use. For staining, free-floating sections were first washed in Tris buffer (50 mM, pH 7.4) with 0.05% Triton X-100 (X100, Sigma-Aldrich) and blocked at room temperature (RT) for 1 h in Tris buffer containing 0.3% Triton X-100 and 5% donkey serum. Sections were then incubated overnight at 4 °C with primary antibodies (Table 1). The

next day, sections were washed in Tris buffer and incubated for 1 h at RT with Alexa Fluor 488-, Cy3- and/or Cy5-conjugated secondary antibodies. For nuclei labeling, sections were incubated for 10 min in Tris-Triton 0.05% with DAPI (1:10,000, AB228549, Abcam). Sections were mounted on SuperFrost Plus slides (J1800AMNZ, Epredia) in Dako Fluorescence Mounting Medium (S3023, Dako).

Imaging was performed using a Zeiss LSM 800 or LSM 900 confocal laser scanning microscope with 10× (Plan-Apochromat, NA 0.45), 25× (LCI Plan-Neofluar, NA 0.8), or 40× objectives (Plan-Apochromat, NA 1.4, Oil DIC (UV) VIS-IR). Overview images were taken with the 10× objective, and tiled images were aligned using ZEN software (Zeiss) with a 5% overlap to minimize tiling artifacts. Z-stacks were acquired at 1 μm step size, with 10 slices per stack. For image analysis, ImageJ (Fiji, Version 2.0.0-rc-69/1.52) was used. Maximum intensity projections were analyzed. Fluorescent particle area was measured for GFAP and IBA1, and cell counts were performed using S100β and NeuN. DAPI staining ensured comparable brain area selection. For Sholl analysis, maximum intensity projections were despeckled and binarized, then analyzed with the Neuroanatomy Plugin (Legacy: Sholl analysis) in ImageJ. The first concentric circle was set at 4 μm for astrocytes and microglia, with subsequent 2 μm radius steps. For each animal, three brain sections and at least three images per section were analyzed.

## RNA isolation and translatome analysis

To isolate actively translated RNA, the translating ribosome affinity purification (TRAP) protocol was followed as previously described[79]. Briefly, cortex samples were homogenized with 1 ml lysis buffer (20 mM HEPES KOH (pH 7.3), 150 mM KCl, 10 mM MgCl2, 0.5 mM DTT, 100 μg/ml cycloheximide, 10 μl/ml rRNasin and 10 μl/ml Superasin, EDTA-free protease inhibitors) in a tissue grinder (3432S90, Thomas Scientific) on a rotor (LT-400D, Yamato) with 12 strokes. The lysate was centrifuged for 10 min at 2000 × g at 4 °C. 110 μl 10% NP-40 and 120 μl 300 mM DHPC were added to the supernatant before centrifuging for 10 min at 17,000 × g at 4 °C. The supernatant was transferred to a fresh tube for immunepurification (IP). 50 μl of each sample was kept at 4 °C as an input (pre-IP) sample until RNA elution and purification. The affinity matrix was prepared as recommended, and optimal bead titers for this approach were determined using an affinity matrix titration beforehand. A total volume of 200 μl affinity matrix containing 75 μl

Streptavidin MyOne T1 Dynabeads (65601, Invitrogen), 30 µl protein L, and 12.5 µl of each GFP antibodies 19C8 and 19F7 was added to each sample for IP. All samples were incubated on a tube rotator for 17 hours at 4 °C. Then, the samples were washed four times with high-salt buffer (20 mM HEPES KOH (pH 7.3), 350 mM KCl, 10 mM MgCl2, 1% NP-40, 0.5 mM DTT and 100 µg/ml cycloheximide). RNA elution and purification were performed using the RNeasy Plus Micro Kit (74034, Qiagen) for both IP and input samples at RT. Briefly, RNA was dissociated from ribosomes by adding 350 µl RLT buffer containing 40 µM DTT and further purified following the manufacturer's instructions. RNA concentration was measured using the Quantifluor RNA System (E3310, Promega) and RNA integrity was assessed using the RNA 6000 Pico Kit (5067-1513, Agilent Technologies) on a 2100 Bioanalyzer (Agilent Technologies Inc., CA).

For real time quantitative PCR (qPCR) analysis, QuantiTect Reverse Transcription Kit (205311, Qiagen) was used for cDNA synthesis, and qPCRs were formed using SYBR green (04887352001, Roche) on a CFX384 Touch Real-Time PCR Detection System (Bio-Rad). The following primers used for expression analysis: for *Gfap* (5′-TGGCCACCAGTAACATGCAAGAG-3′, 5′-CGTCTGTGAGGTCTGCAAACTTAG-3′), for *Cyc1* (5′-TGCTACACGGAGGAAGAAGC-3′, 5′CCATCATCATTAGGGCCATC-3′), and for *Slc2a1* (5′-TGTCGGGTATCAATGCTGTGT-3′, 5′-GATACCGGAGCCGATGGTG-3′).

For RNA-sequencing analysis, the cDNA and generation of libraries were performed with the Smart-seq2 protocol[80]. Single-end sequencing (100nt) was performed at the Functional Genomics Center Zurich (FGCZ) core facility with an Illumina Novaseq 6000. Samples were each run on two lanes and demultiplexed. A sequencing depth of ~20 M reads per sample was used. Adapters were trimmed using cutadapt[81] with a maximum error rate of 0.05 and a minimum length of 15. The coverage of *Slc2a1* exons was based on read counts in a STAR alignment; for all other purposes kallisto[82] was used for pseudo alignment of reads on the transcriptome level using the genecode.vM17 assembly with 30 bootstrap samples and an estimated fragment length of 200 ± 20. One sample was excluded due to a very low read count, and IP samples that showed a high expression of neuronal genes were excluded. For differential gene expression (DGE) analysis we aggregated reads of protein coding transcripts and used R (v. 3.6.2) with the package "edgeR" (v 3.26.8) for analysis. A filter was used to remove genes with low expression prior to DGE analysis. edgeR was then used to calculate the normalization factors (TMM method) and estimate the dispersion (by weighted likelihood empirical Bayes). For two group comparisons the genewise exact test was used. For multiple testing correction the Benjamini–Hochberg false discovery rate (FDR) method was used. GO enrichment analysis was done with topGO, using Fisher's test and the weight01 algorithm. Heatmaps were produced with the sechm package. To avoid rare extreme values from driving the scale, the color scale is linear for values within a 98% interval, and ordinal for values outside it. Unless otherwise specified, the rows were sorted using the features' angle on a two-dimensional projection of the plotted values, as implemented in sechm.

## Behavioral analysis

Experiments were conducted in adult male mice 60 days posttamoxifen treatment. Mice were handled daily for 20 min over two weeks before behavioral testing, which took place during their dark (active) phase. The experimenter was blinded to genotype for all behavioral assessments.

Sensorimotor performance (scoring) was assessed using gait assessment, ledge test, hindlimb clasping, and horizontal wire tests (3 mm and 6 mm diameter). Each test was scored 0–3, except for the horizontal wire tests (0–4), with a maximum total score of 17, where higher scores indicate greater sensorimotor deficits. Gait assessment was performed on a table with the mouse facing away from the experimenter. A score of 0 was assigned for normal movement, where all limbs supported body weight, no tremor was observed, and the abdomen did not touch the floor. A score of 1 indicated light tremor, while a score of 2 reflected an uncoordinated, halting walk with feet pointing outward. A score of 3 indicated severe tremor, hesitant movement, abdomen dragging, and the tail aligned along the surface. In the ledge test, mice were placed on the edge of a cage starting with their forepaws, and their ability to walk was assessed. A score of 0 was assigned if the mouse walked in a coordinated manner, maintained balance, and descended using its forepaws. A score of 1 reflected coordinated walking with minor slips, while a score of 2 was given if the mouse exhibited frequent hindlimb slips and mild tremor. A score of 3 indicated the mouse fell off the ledge due to hindlimb placement failure, showed severe tremor, and landed on its head. Hindlimb clasping was evaluated by lifting the mouse by the tail 30–40 cm above a table for 10 s. A score of 0 indicated that the hindlimbs were fully extended. A score of 1 was given if one hindlimb was partially retracted for at least 5 s, while a score of 2 indicated that both hindlimbs were partially retracted for at least 5 s. A score of 3 was assigned if both hindlimbs were fully retracted for more than 5 s. Horizontal wire tests were performed by placing the mouse with its forepaws on the center of a wire 30 cm above a cushioned surface, and the latency to fall was recorded. A score of 0 was assigned if the mouse reached the platform or fell after ≥30 s, while scores of 1, 2, 3, and 4 were assigned for falling between 21–30 s, 11–20 s, 6–10 s, and 1–5 s, respectively. Each test was performed twice per day for two consecutive days (four trials total). The final score was calculated by averaging the four trial scores for each test and summing these averages across all tests.

The Barnes maze test was used to assess spatial learning and memory, as previously described[40]. The custom-made Barnes maze table (105 cm diameter) had 22 holes (5 cm diameter, 6 cm apart), all closed from underneath except for the target hole, which had an escape box below it, allowing the mouse to hide inside. A weak aversive stimulus (fan, light, and buzzer) was used to motivate the mouse to locate the target hole. Visual cues in the room and around the table remained constant to help the mouse orient itself and learn the target hole location. Each session was recorded using a GoPro HERO4 camera. To ensure a standardized learning experience across all mice, a gentle guidance and familiarization process was implemented. During the first training sessions on adaptation day 0, if a mouse failed to locate the target hole within the allotted time, the experimenter gently guided it toward the hole. This brief guidance was applied uniformly to all animals, ensuring that each mouse had comparable exposure to the maze and target location while still allowing for individual learning. The spatial acquisition phase (days 1–4) consisted of three sessions per day (1 h apart). Each session began with the mouse placed in the center of the table under a dark box. After 10 s, the box was lifted, aversive stimuli were activated, and latency to locate the target hole was measured. Once the mouse found the target hole, it was allowed to remain in the escape box for 2 min to rest and reinforce memory formation. Sessions lasted a maximum of 180 s. On test day (day 5), the target hole was closed like all other holes, preventing the mouse from entering it. 90-sec video recordings were analyzed using DeepLabCut (2.1.8.1) and R (3.6.1). A model was trained to track the nose, ears, and tail base of the mouse in each video frame. These markers were mapped against ROIs for each hole in R, enabling quantification of nose-poke counts, latency to locate the former target hole, and distance walked.

The passive avoidance test was used to assess fear-motivated learning and memory. The test chamber consisted of a light compartment and a dark compartment, separated by a door. During the acquisition phase, the mouse was placed in the light compartment and allowed to explore for 120 s. The door then opened, and the latency to cross into the dark compartment was recorded. Upon entering the

dark compartment, the door closed, and a mild foot shock (0.4 mA, 2 s) was delivered through the floor grid. The mouse remained in the dark compartment for 30 s to allow association between the dark compartment and the aversive stimulus. During the test phase (1 h and 24 hrs post-acquisition), the mouse was again placed in the light compartment for 120 s of exploration. Once the door opened, the latency to cross into the dark compartment was measured. The test ended if the mouse did not cross within 600 s.

## AAV injections

7- to 9-week-old mice were anesthetized with midazolam (5 mg/kg), fentanyl (0.05 mg/kg), and medetomidine (0.5 mg/kg) and secured in a stereotactic frame. Craniotomies (1 mm diameter) were made above the left somatosensory cortex using a high-speed drill. A glass micro-pipette (5-000-1001-x, PCR Micropipettes, Drummond) and a custom-made microinjector pump were used to inject 200 nL of viral mix at 400 µm depth at 50 nL/min. The viral mix contained AAV2/8-CAG-DIO-TdTomato ($5.5 \times 10^{12}$ vg/mL, Gene Therapy Center Virus Vector Core Facility, University of North Carolina) and the FRET-based glucose sensor[41] AAV2/9-hGFAP-FLII12Pglu-700µΔ6 (termed FLIIP, $5.7 \times 10^{12}$ vg/mL, Viral Vector Core Facility, University of Zurich). The FLIIP construct was codon-diversified to prevent homologous recombination[83]. After surgery, mice received an antagonist solution (flumazenil, 0.5 mg/kg; atipamezole, 2.5 mg/kg) and buprenorphine (0.1 mg/kg s.c.) for analgesia. Mice were monitored carefully to ensure proper recovery. One week post-injection, tamoxifen was administered as described above to induce DIO-TdTomato expression and GLUT1 recombination.

## Acute brain slice preparation

Mice were deeply anesthetized with isoflurane and after decapitation, brains were quickly dissected in an ice-cold ACSF (in mM: 125 NaCl, 2.5 KCl, 25 NaHCO₃, 1.25 NaH₂PO₄, 1 MgCl₂, 2 CaCl₂, 25 glucose, pH 7.4). Coronal sections (350 µm thick) were cut in cutting solution (in mM: 65 NaCl, 2.5 KCl, 25 NaHCO₃, 1.25 NaH₂PO₄, 7 MgCl₂, 0.5 CaCl₂, 105 sucrose, 25 glucose, pH 7.4) using a vibratome (Vibration micro-tome, HM 650 V, VWR) and afterwards kept in ACSF at 34 °C for half an hour and then at room temperature until imaged. All solutions were continuously oxygenated with 95% O₂ and 5% CO₂.

## Metabolic imaging and analysis

Slices were transferred to a recording chamber (Luigs & Neumann LN insert with a drill hole for perfusion, mounted with a cover glass) and continuously perfused with oxygenated ACSF (95% O₂, 5% CO₂) containing (in mM: 125 NaCl, 2.5 KCl, 25 NaHCO₃, 1.25 NaH₃PO₄, 1 MgCl₂, 2 CaCl₂, 3 lactate, 5 glucose, 22 sucrose, pH 7.4). In glucose uptake and aglycemia experiments, glucose was replaced with sucrose to maintain osmolarity. Temperature was maintained at 33 °C using an in-line solution heater (ALA HPC-G, Scientific Instruments). Slices equilibrated for 40 min in the recording chamber before experiments. Astrocytes were imaged using a custom-built two-photon scanning laser microscope[84] with a tunable pulsed laser (MaiTai eHP DS system, Spectra-Physics) and a 20× water immersion objective (W Plan-Apochromat 20×/1.0 DIC VIS-IR, Zeiss). Unidirectional frame scans were acquired at 256 × 256 pixel resolution with a 6.4-µs pixel dwell time using ScanImage (r3.8.1). The region of interest (ROI) was selected based on an overview Z-stack acquired at 1 µm step size and taken at 1070 nm (for TdTomato-expressing astrocytes) and 870 nm (for FLIIP-expressing astrocytes). Donor and acceptor fluorescence signals were simultaneously collected using photomultiplier tubes, a 560-nm edge dichroic beam splitter (BrightLine, Semrock), and band-pass filters of 545/55 nm (yellow channel) and 475/50 nm (blue channel). ROIs typically contained 5–8 co-expressing cells. Experiments typically ended with an aglycemia protocol for one-point calibration of the sensor. For glycolysis measurements, cytochalasin B (20 µM, 5474, Tocris) was

used to block glucose transporters as previously described[42]. For glucose uptake in decoupled astrocytes, the gap-junction blocker carbenoxolone (150 µM, C4790, Sigma) was applied. Single-cell analysis was performed using a custom MATLAB script (available at https://gitlab.com/einlabzurich/fretanalysis). Images were motion-corrected using a correlation-based algorithm and time-smoothed with a boxcar kernel. The smoothing window was optimized by testing progressively larger windows to avoid temporal distortion (final window size: 11 frames). The YFP/CFP ratio was calculated per ROI (single cell) to assess astrocytic glucose changes. To minimize numerical errors, ratios were calculated by first averaging all pixels in a given channel and then performing division, rather than using pixel-by-pixel division followed by averaging.

## Thrombin stroke model and analysis

We induced a focal cerebral ischemia using the thrombin-model of stroke in male mice 60 days post-tamoxifen treatment. The procedure steps for the cerebral ischemia induction were performed as described previously[48]. In brief, mice heads were fixed in a stereotactic frame, the skin between the left eye and ear was incised and the left temporal muscle retracted. After craniotomy above the middle cerebral artery (MCA) M2 segment, the dura was removed, and a glass pipette (calibrated at 15 mm/µl; Assistant ref. 555/5; Hoechst) was introduced into the lumen of the MCA. Through the pipette, 1 µl of purified human alpha-thrombin (1UI; HCT-0020, Haematologic Technologies Inc., USA) was injected to induce the formation of a clot in situ. Ten minutes after thrombin injection, the pipette was removed. Ischemia induction was considered successful when cortical cerebral blood flow (CBF) rapidly dropped to at least 50% of baseline level in the MCA territory. CBF changes were monitored before and during ischemia, using a laser speckle contrast imaging monitor (FLPI, Moor Instruments). The acquisition was performed with a frame rate of 0.25 Hz. LSI images were generated with arbitrary units in a 256-color palette by the MOOR-FLPI software.

Seven days after stroke induction, mice were euthanized by an overdose i.p. injection of pentobarbital (200 mg/kg) followed by perfusion and tissue preparation for immunohistochemistry as detailed above. Coronal sections were stained for DAPI (nuclei), NeuN (neurons), and GFAP (astrocytes). Lesion volume was quantified using both NeuN signal loss and GFAP labeling, which clearly delineated the lesion border through reactive gliosis. The infarcted area was measured on each slice using ImageJ and then multiplied by the ratio of the surface of the infarcted cortex (ipsilateral) to the intact cortex (contralateral) to correct the lesion for brain edema. Infarct volume was calculated by linear integration of infarcted areas in 6 coronal sections for each animal.

## Statistics

Statistical analysis was performed using GraphPad Prism and R (version 3.6.1). Intergroup comparisons were conducted using two-sided paired or unpaired t-tests, as specified in the figure legends. For analyses involving multiple comparisons, one-way or two-way ANOVA was applied. Datasets had to pass the Shapiro–Wilk test for normality before being subjected to parametric tests (t-test and ANOVA); otherwise, nonparametric tests were used to determine statistical significance. Equal variances were assumed but not tested. Biosensor imaging data, Sholl analysis, and infarct volume measurements were analyzed using linear mixed-effects models, accounting for cells or stainings nested within distinct brain slices and animals. Significance levels were defined as follows: $*p < 0.05$, $**p < 0.01$, $***p < 0.001$. P values and sample sizes are stated in the figure legends. Sample sizes were not pre-determined but were constrained by the availability of age-matched transgenic mouse cohorts. Where applicable, experimenters were blinded to genotype. This study was not designed to investigate sex-specific effects; animals were used as available, and

post hoc sex-based analyses were not performed due to insufficient sample sizes for meaningful statistical conclusions. Data are presented as individual values with mean ± SEM, or as box plots displaying the median (center line), quartiles (box bounds), mean (+) and min-to-max or 1.5× interquartile range (whiskers), as indicated in each figure legend.

## Reporting summary
Further information on research design is available in the Nature Portfolio Reporting Summary linked to this article.

## Data availability
All data supporting the findings of this study are included in the paper and its Supplementary Information. RNA-sequencing data have been deposited at Gene Expression Omnibus (GEO) under the accession code GSE223687. Additional information related to this study is available from the corresponding authors. Source data are provided with this paper.

## Code availability
The code used for FRET image analysis is available at https://gitlab.com/einlabzurich/fretanalysis.

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

## Acknowledgements

We thank Noemi Binini and Kim David Ferrari for technical assistance and all laboratory members for frequent discussions and critical input, Kathrin Kusch for kindly providing the GLUT1 antibody, J.-C. Paterna and the viral vector facility of the Neuroscience Center Zurich for AAV production, and the Functional Genomics Center Zurich for RNA-seq support. S.W. received support from the Swiss National Science Foundation (SNSF; 310030_200703). L.F.B. was supported by Fondecyt 1230145. M.E.A. was supported by the Swiss Heart Foundation (FF21068) and Vontobel Foundation (1104/2024). B.W. was supported by the SNSF (Grant no. 219656). A.S.S. was supported the Cloëtta Foundation and the SNSF (Eccellenza PCEFP3_187000 and 10003348).

## Author contributions

L.T., H.S.Z, J.D., U.D., M.T.W., R.W., Z.J.L., M.E.A. and A.S.S performed experiments and analyzed data. P.-L.G. and L.M.vZ. conducted RNA-seq data analysis, L.H., L.R., and M.E.A. contributed to supervision and data analysis. E.D.A., J.B., S.W., L.F.B., M.E.A., B.W. and A.S.S. provided resources and contributed to experimental design. A.S.S conceptualized and supervised the study and wrote the manuscript with input from all authors.

## Competing interests

The authors declare no competing interests.
