## [Transparent Peer Review file · Nature Communications]

Astrocytic GLUT1 deletion in adult mice enhances glucose metabolism and resilience to stroke

Corresponding Author: Professor Aiman Saab

Version 0:

Reviewer comments:

Reviewer #1

(Remarks to the Author)

In the manuscript "Astrocytic GLUT1 deletion in adult mice 1 enhances glucose metabolism and resilience to stroke" authors studied the effects of eliminating the glucose transport GLUT1 in astrocytes via inducible conditional KO under physiological and stroke conditions. They report that GLUT1 cKO astrocytes increase their glucose metabolism which provides certain degree of neuroprotection in a model of stroke by limiting the size of the injury.

This manuscript is well written, data is very nicely presented, and it was easy to follow the authors' rationale. Experiments were carried out and analyzed rigorously. The only major comment refers to the stroke part of the study, which feels underdeveloped. Some comments are provided in order to extend and improve that part.

1. Figure 1e and f. GS may not be the best astrocyte marker. Additional staining to prove the efficiency of the cKO would be desirable. S100B as used in subsequent experiments would be a good choice. Authors should also consider performing some sort of quantification to assess the Glut1 protein reduction in astrocytes, specifically, as the only quantification they show is the Western blot with brain homogenates.
2. Sup Fig 3c, d, e and f: a lower magnification image to indicate where the ROIs considered for analyses are located will be useful to the reader.
3. NeuN staining is not an appropriate marker to measure injury size. Authors should perform TTC stainings, and/or hematoxylin and eosin (H&E) stainings and analyze injury size from there.
4. Since their studies are focused on astrocytes, it would be interesting to analyze astrocyte response in stroke when GLUT1 is ablated. It is known that GLUT1 is increased in response to ischemia, and it was suggested that GFAP+ astrocytes may be the ones upregulating GLUT1 in stroke. It will be interesting in this study to analyze potential GFAP changes in the glut1 cKO at day 7 after lesion (also comparing to sham animals).
5. If Glut1 cKO mice are, in fact, somehow protected from stroke, authors could perform the behavioral tests they did in the non-injured animals to assess whether Glut1 cKO perform better compared to Ctrl animals, after thrombin-induced stroke.
6. Authors used males and females in their study – however, in the behavioral tests they state they only used males. Can authors specify which experiments used data from both sexes? Sex differences are reported in stroke, and it may be beneficial to separate the results by sex in their stroke-related experiments.
7. In the Barnes maze protocol, authors state that "if the mouse did not find the target hole, the experimenter gently guided it towards it." Was this something that had to be performed often? Was it more often in one genotype over the other? If one genotype was moving/exploring less than the other, hence the need to "guide it", it could be a phenotype. If only a small number of animals had to be "guided", it'd be recommended to eliminate those mice from the analyses and it may introduce bias in the results.

Minor comments:

1. In the methods section, please add catalog numbers where possible.
2. In the imaging section, please specify the parameters of the z-stacks (how many steps, measure, overlap?)

Reviewer #2

(Remarks to the Author)

Overall, I find this study interesting and with potential relevance to brain functions. The paper is well written and nicely

illustrated.

One fundamental shortcoming from the perspective of this reviewer is that region-specific Glut-1 KO was not accomplished. The results are not consistent with some other works that altered astrocytes region-specifically and/or broadly with very specific phenotypes, including systemic metabolism. I would like the authors to discuss the issues.

Reviewer #3

(Remarks to the Author)

In this manuscript the authors investigate the role of astrocytic glucose transporter GLUT1 in astrocytic glucose metabolism as well as its contribution to brain homeostasis and responses to perturbation.

Using a combination of genetic, viral, and pharmacological manipulations both in slice and in vivo, the authors show that astrocyte-specific GLUT1 is dispensable to normal brain function (Figure 3) and behavior (Figure 2). Astrocytic deletion of GLUT1, the primary glucose transporter of this cell type, leads to no significant difference in basal glucose levels or in glucose uptake kinetics (Figure 3). However, there was a surprisingly significant increase in intracellular glucose levels in GLUT1-deleted astrocytes without compensation from other glucose transporters or connexin hemichannels (Figures 1, 4, S1, S2, and S6). The authors also posit that increased glucose metabolism in GLUT1-deleted astrocytes may be neuroprotective, as GLUT1 conditional knockout mice exhibit smaller infarct volumes following a model of stroke (Figure 5).

There are a few minor points that may be worth addressing:

- TRAP RNAseq. The authors write that there are only moderate differences in gene regulation between the genotypes as well as no significant changes related to glucose or monocarboxylate transport (Figure S2). The data presented in this figure – especially in the heatmap showing the top 13 differentially (spelled as “deferentially” in the manuscript) regulated genes are not strong support for these claims. I would recommend addition of genes related to glucose and monocarboxylate transport to the heatmap, with a note on the exact number of differentially regulated genes in either the main or figure text to quantify what the authors consider “moderate” gene differences. Alternatively, both up- and down-regulated DEGs could be plotted on a volcano plot (showing log₂FC and significance values) to show numbers of both significant and nonsignificantly changed genes.

- The concluding remarks on the manuscript’s behavioral data. The authors suggest that “neurotransmission and synaptic plasticity likely remain intact despite loss of astrocytic GLUT1.” While not a definite statement, it is still a bold one. Perhaps the authors could also query genes related to astrocyte synaptogenesis/long-term plasticity in the TRAP RNAseq? Alternatively, this comment could be removed or tempered – even without the speculation on neurotransmission and synaptic plasticity, the behavior results remain interesting.

- The use of Slc1a3 as an astrocyte-specific driver. Like most astrocyte-enriched targeting, Slc1a3 also is expressed in neural progenitors that give rise to neurons in the olfactory bulb and hippocampus, and only targets a small fraction of all astrocytes (discussed by the authors in this manuscript’s Discussion and reviewed nicely by PMID: 32042146). Have the authors taken advantage of this potential sparse labeling and compared recombined and nonrecombined astrocytes within the same animal?

- Gap junctions. The authors note in the discussion that inhibition of gap junctions and hemichannels led to a decrease in glucose levels, and that connexin-mediated regulation of astrocyte glucose metabolism remains unclear. Connexin-dependent astrocytic glucose transport across long distances has previously been reported in a mouse model of glaucoma (PMID: 32690710). Could it be that basal glucose levels are comparable in the GLUT1 cKO mice by compensation from nonrecombined astrocytes? It does seem like the distribution of values for individual cells (for example, in Figure 3e) is larger, perhaps reflecting two populations of recombined and nonrecombined cells? Additionally, the stroke lesion reduction experiment is interesting – are there intrinsic differences between how recombined and nonrecombined astrocytes work with or respond to glucose post-stroke?

Ultimately, these experiments support the idea that astrocytes are highly metabolically plastic in their ability to support proper neural function. The presented work in this manuscript has the potential to be interesting to groups studying the nervous system in both health and disease, and spans multiple levels of analysis.

Reviewer #4

(Remarks to the Author)

The primary objective of the study by Thieren and colleagues was to determine the functional role of the glucose transporter Glut1 expressed by astrocytes. The authors present molecular and functional analyses performed in conditional mutant mice for astrocytic Glut1 and report intriguing findings including the preservation of sensorimotor and cognitive performances, an increase in glucose consumption by astrocytes as well as neuroprotection during an ischemic episode.

These data are unexpected as the 45 kd glut1 is thought to be the main (if not unique) glucose transporter expressed by astrocytes. The question of the membrane transporter through which glucose enters the cells of these KO mice remains enigmatic since the authors have shown that there is no compensation for the expression of other known transporters. Overall, I consider that the study was carried out rigorously by the authors and that, in particular, the validation of the cell specific glut1-deficient mouse model is convincing.

I do, however, have a number of points and questions which, if they can be addressed by the authors, could increase the overall impact of their study.

1. In vivo evidence that glucose consumption is increased. The study of the mechanisms regulating cerebral glucose metabolism is particularly relevant in vivo because it ensures that all the steps from blood glucose supply to local distribution and use are taken into account, including all the molecular and cellular players involved. It therefore seems to me that the current study would be greatly improved by the in vivo measurement of glucose consumption by glut1 KO mice. It would also be informative to measure the blood glucose levels of these animals. Such data would be very complementary to the one already obtained using FRET sensors in slices.

2. Cellular expression of the protein Glut1. The authors could consider using immunogold electron microscopy to convincingly demonstrate the loss of Glut1 expression in astrocyte membranes and the potential relocation of residual transporters.

3. Glycogen content. Glycogen is stored in astrocytes and is thought to provide lactate as an energy source to neurons through monocarboxylate transporters (MCTs) to maintain memory formation. It would be of particular interest to determine whether glycogen levels are altered in glut1 KO mice.

Minor points.

1. The sentence starting line 67 "The structure and function...also due to its implication" should be reworded.

2. Figure 4. The rates of glucose rise (Fig. 4b) as well as the glucose changes (Fig. 4c) vary greatly between cells. Did the authors use the number of cells (n= 88 and 55) for their statistics or the number of experiments?

3. Suppl. Fig.3. I do not quite understand how semi-quantification was achieved using GFAP, S100beta, IBA1 and NeuN immunostaining using a very low number of animals.

Version 1:

Reviewer comments:

Reviewer #1

(Remarks to the Author)

Authors have addressed all my comments satisfactorily and I don't have any further remarks. I look forward to seeing this beautiful study published.

Reviewer #2

(Remarks to the Author)

In my view the authors adequately addressed the comments raised by the 4 reviewers.

Reviewer #3

(Remarks to the Author)

In this manuscript the authors investigate the role of astrocytic glucose transporter GLUT1 in astrocytic glucose metabolism as well as its contribution to brain homeostasis and responses to perturbation. Using a combination of genetic, viral, and pharmacological manipulations both in slice and in vivo, the authors show that astrocyte-specific GLUT1 is dispensable to normal brain function (Figure 3) and behavior (Figure 2). Astrocytic deletion of GLUT1, the primary glucose transporter of this cell type, leads to no significant difference in basal glucose levels or in glucose uptake kinetics (Figure 3). However, there was a surprisingly significant increase in intracellular glucose levels in GLUT1-deleted astrocytes without compensation from other glucose transporters or connexin hemichannels (Figures 1, 4, S1, S2, and S6). The authors also posit that increased glucose metabolism in GLUT1-deleted astrocytes may be neuroprotective, as GLUT1 conditional knockout mice exhibit smaller infarct volumes following a model of stroke (Figure 5). Ultimately, these experiments support the idea that astrocytes are highly metabolically plastic in their ability to support proper neural function. The presented work in this manuscript has the potential to be interesting to groups studying the nervous system in both health and disease, and spans multiple levels of analysis.

The authors have addressed my concerns through this revision sufficiently. I have no further comments.

Thieren et al., NCOMMS-24-42691 Point-to-point response

The reviewers' comments are cited directly from the decision letter. Below are the individual comments (black) and our responses (blue). Of note, the figure referencing in this reply refers to the updated figure numbering in the revised version of the manuscript

Reviewer #1 (Remarks to the Author):

In the manuscript “Astrocytic GLUT1 deletion in adult mice 1 enhances glucose metabolism and resilience to stroke” authors studied the effects of eliminating the glucose transport GLUT1 in astrocytes via inducible conditional KO under physiological and stroke conditions. They report that GLUT1 cKO astrocytes increase their glucose metabolism which provides certain degree of neuroprotection in a model of stroke by limiting the size of the injury.

This manuscript is well written, data is very nicely presented, and it was easy to follow the authors' rationale. Experiments were carried out and analyzed rigorously. The only major comment refers to the stroke part of the study, which feels underdeveloped. Some comments are provided in order to extend and improve that part.

Response: We thank the reviewer for the positive feedback. As suggested by the reviewer, we have now added more analysis to improve the stroke part of our study.

1. Figure 1e and f. GS may not be the best astrocyte marker. Additional staining to prove the efficiency of the cKO would be desirable. S100B as used in subsequent experiments would be a good choice. Authors should also consider performing some sort of quantification to assess the Glut1 protein reduction in astrocytes, specifically, as the only quantification they show is the Western blot with brain homogenates.

Response: We thank the reviewer for this thoughtful suggestion. We have demonstrated a significant 50% reduction in GLUT1 protein levels via Western blot in brain homogenates, which include all cell types. To specifically assess astrocytic GLUT1 deletion as well as to determine the efficacy of astrocytic deletion, we performed both astrocyte-specific RNA-Seq analysis (of translating mRNAs using TRAP) and qPCR, both of which confirmed a ~70% reduction in *Slc2a1* transcript levels in our cKO mice (Fig. 1j, 1k). These complementary approaches confirm the efficacy of the astrocyte-specific GLUT1 deletion. Using immunohistochemistry, we further validated the deletion of GLUT1 specifically from astrocytes. While we acknowledge that GS is not the only marker available for astrocytes, it remains a well-established and widely used marker, especially for grey matter astrocytes. Importantly, GS allows visualization of both astrocyte somata and their primary processes, similar to S100 β staining, as seen in the example image of GS and S100 β co-labelling we provide here solely for this response letter.

Notably, combining S100 β with GLUT1 immunostaining was not feasible due to both antibodies being derived from the same host species (rabbit). We were unable to find and obtain a non-rabbit anti-S100 β antibody reactive to mouse S100 β ; as most alternatives are specific to human S100 β .

However, we are convinced of the reliability of GS as a robust astrocyte marker, with which we could validate the astrocyte-specific GLUT1 deletion in our cKO mice. And we hope the reviewer agrees that the GS immunolabeling used in our study is sufficient to validate the GLUT1 deletion from astrocytes in our cKO mice. And with both Western blot and astrocyte-specific mRNA analysis we reliably showcase clear depletion of GLUT1 in our cKO mice.

2. Sup Fig 3c, d, e and f: a lower magnification image to indicate where the ROIs considered for analyses are located will be useful to the reader.

Response: As suggested, we have now added an exemplary lower magnification image to indicate the brain region used for analyses in the supplementary figure (**NEW Supplementary Fig. 3c**).

3. NeuN staining is not an appropriate marker to measure injury size. Authors should perform TTC stainings, and/or hematoxylin and eosin (H&E) stainings and analyze injury size from there.

Response: We appreciate the reviewer's valuable feedback and recognize that NeuN staining is not the classical method for measuring injury size in stroke. Traditional histological methods, such as 2,3,5-triphenyltetrazolium chloride (TTC) staining and hematoxylin and eosin (H&E) staining, are widely used for lesion assessment. However, studies show that TTC staining is reliable for acute stroke lesion measurements (up to seven days post-stroke) but is not suitable for long-term assessments. Alternatively, immunolabeling of fixed tissue offers the advantage of higher cellular resolution, allowing for more precise delineation of the affected area. By seven days post-stroke, the glial scar provides a structural marker that can be used to define the lesion border. To further strengthen our findings, we have now included an additional analysis using GFAP labeling, which clearly delineates the lesion borders through reactive gliosis. Quantification of injury volume using both NeuN loss and GFAP labeling consistently demonstrated a reduction in injury size in our cKO mice (**NEW Fig. 5e-f**).

We believe that combining these two markers provides a robust and well-validated approach for accurately assessing stroke-induced injury. We hope this additional analysis sufficiently addresses the reviewer's concern.

(References: PMID: 38219841, PMID: 29401502, PMID: 14671235, PMID: 30679481)

4. Since their studies are focused on astrocytes, it would be interesting to analyze astrocyte response in stroke when GLUT1 is ablated. It is known that GLUT1 is increased in response to ischemia, and it was suggested that GFAP+ astrocytes may be the ones upregulating GLUT1 in stroke. It will be interesting in this study to analyze potential GFAP changes in the glut1 cKO at day 7 after lesion (also comparing to sham animals).

Response: We thank the reviewer for this suggestion. We have now analyzed GFAP+ astrocyte morphology at both the infarct border and the peri-infarct region. The infarct border was defined as the region extending 0-150 μm from the infarct edge, characterized by astrocytes exhibiting an elongated morphology oriented toward the infarct core. The peri-infarct region was classified as the adjacent area 150-500 μm beyond the infarct border, where astrocytes remained reactive but showed a less elongated morphology. Sholl analysis revealed that while infarct border astrocytes in both genotypes displayed a similar elongated morphology toward the lesion, peri-infarct astrocytes in cKO mice exhibited a less reactive profile compared to littermate controls (**NEW Fig. 5g-i**). Additionally, microglial morphology in the peri-infarct region was comparable between genotypes (**NEW Supplementary Fig. 7**).

These findings suggest that the reduced lesion size in cKO mice is associated with a dampened astrocytic response in the peri-infarct region, potentially contributing to limiting stroke size and promoting neuroprotection. This supports the idea that GLUT1-deficient astrocytes, with their enhanced glucose metabolism, may exert a more neuroprotective role following stroke. To confirm that this effect is stroke-specific, we also examined astrocyte morphology in naïve mice that were not subjected to stroke. In this condition, cortical astrocytes displayed comparable morphology across genotypes (**NEW Supplementary Fig. 3e**), indicating that the reduced astrocytic reactivity in cKO mice occurs specifically in response to stroke.

5. If Glut1 cKO mice are, in fact, somehow protected from stroke, authors could perform the behavioral tests they did in the non-injured animals to assess whether Glut1 cKO perform better compared to Ctrl animals, after thrombin-induced stroke.

Response: We appreciate the reviewer's suggestion and recognize the importance of assessing functional recovery following stroke. In fact, we had conducted behavioral tests both before and 7 days after thrombin-induced stroke to determine whether male GLUT1 cKO mice perform better than Ctrl animals. We assessed neurological deficits using the neurological score and sticky tape test, which, based on our experience and prior studies, are among the most sensitive behavioral assessments for the thrombin stroke model in mice, given its small, localized infarction.

Notably, **even control animals did not exhibit neurological deficits 7 days post-stroke, despite clear histological lesions**. While our results showed no significant differences in performance between genotypes, cKO mice exhibited a slight trend toward improved performance in the adhesive removal test (see **REV. Figure** below, provided solely for this response letter and the reviewer's benefit).

We believe that lack of overt behavioral deficits is primarily due to the small lesion sizes in C57BL/6 mice, which possess extensive collateral networks that mitigate stroke severity (Binder, El Amki et al., 2024). Hence, the limited initial tissue damage does not always translate into overt behavioral impairments, making it difficult to resolve a neuroprotective effect based on behavioral analysis in our model.

We acknowledge that this is a limitation of the thrombin stroke model compared to proximal MCA occlusion models, which induce larger infarcts and more severe neurological deficits. However, we strongly believe that the absence of behavioral differences does not diminish the significance of our histological findings, which provide robust evidence that astrocytic GLUT1 deletion confers neuroprotection following stroke.

REV. Figure (provided solely for this response letter): Behavioral analysis of male cKO (n = 8) and Ctrl (n = 9) littermates before and 7 days post-stroke. The assessments include (a) neurological score and (b, c) sticky tape test, measuring time to contact (b) and time to remove (c) the tape. Both genotypes showed no signs of neurological deficits post-stroke compared to pre-stroke performance. In (a), a neurological score of 8 indicates the absence of deficits. In (b, c), an increase in time to contact or remove the tape would indicate impairment; however, no such deficits were observed in either group.

6. Authors used males and females in their study – however, in the behavioral tests they state they only used males. Can authors specify which experiments used data from both sexes? Sex differences are reported in stroke, and it may be beneficial to separate the results by sex in their stroke-related experiments.

Response: We apologize for the oversight and thank the reviewer for bringing this to our attention. Our behavioral and stroke-related analyses were conducted exclusively in male mice. Given the well-documented influence of sex on stroke outcomes and behavior, as the reviewer correctly points out, we chose to focus on males to minimize variability. We have now explicitly stated this throughout the manuscript, including the abstract.

For molecular, histological, and imaging experiments, animals were included without consideration of sex, as the study was not designed to investigate sex-specific effects. Post hoc sex-based analyses were not performed, as the available sample sizes were insufficient for drawing meaningful conclusions. This information has now been transparently reported in the Methods section, under both the ‘Animals’ and ‘Statistics’ subsections.

7. In the Barnes maze protocol, authors state that “if the mouse did not find the target hole, the experimenter gently guided it towards it.” Was this something that had to be performed often? Was it more often in one genotype over the other? If one genotype was moving/exploring less than the other, hence the need to “guide it”, it could be a phenotype. If only a small number of animals had to be “guided”, it’d be recommended to eliminate those mice from the analyses and it may introduce bias in the results.

Response: We thank the reviewer for this insightful comment. We acknowledge that our original phrasing in the methods section may have been unclear. To clarify, in our experience with the Barnes maze protocol (as established in our previous work, Hösli et al., 2022), all animals typically require gentle guidance during their initial exposure to the maze – specifically, during the first 1-2 training sessions on adaptation day 0 – to locate the target hole. This step is part of the standard familiarization process and does not reflect differences between genotypes. To prevent any misunderstanding, we have now revised the wording in the methods section to more accurately describe this procedure.

Minor comments:

1. In the methods section, please add catalog numbers where possible.

Response: We have now revised the methods to include catalog numbers where possible.

2. In the imaging section, please specify the parameters of the z-stacks (how many steps, measure, overlap?)

Response: We have specified these parameters as requested in the methods section.

Reviewer #2 (Remarks to the Author):

Overall, I find this study interesting and with potential relevance to brain functions. The paper is well written and nicely illustrated.

Response: We thank the reviewer for the positive feedback.

One fundamental shortcoming from the perspective of this reviewer is that region-specific Glut-1 KO was not accomplished. The results are not consistent with some other works that altered astrocytes region-specifically and or broadly with very specific phenotypes, including systemic metabolism. I would like the authors to discuss the issues.

Response: We appreciate the reviewer's comment and the opportunity to clarify our findings in relation to other very recent work. While region-specific GLUT1 knockouts can provide interesting insights, our study was designed to assess the impact of astrocytic GLUT1 deletion on brain metabolism and function. In our previous studies, we have successfully used the GLAST-CreERT2 approach to dissect astrocyte-specific gene functions (Hösli et al., 2022; Saab et al., 2012), and our findings provide strong evidence that GLUT1 deletion in astrocytes does not critically impair neural health or behavioral performance, despite GLUT1 being traditionally considered the primary glucose transporter in astrocytes.

A very recent study by Ardanaz et al. (2024) also investigated astrocytic GLUT1 deletion using a similar genetic strategy with GFAP-CreERT2. Like us, they observed no major neurological deficits and no apparent upregulation by other glucose transporters. Furthermore, their **18F-FDG PET imaging** revealed a significant increase in whole-brain glucose metabolism in GLUT1-deficient mice. This observation aligns with our findings using biosensor imaging, which shows a **2.6-fold increase in astrocytic glucose uptake and metabolism**. Despite this overall similarity, a key discrepancy arises from their metabolic flux analysis using [U-13C]glucose infusion, where they reported reduced labeled glucose incorporation into astrocytes, interpreting this as a **decrease in astrocytic glucose metabolism**.

Several methodological aspects complicate this interpretation. The **prolonged 3-hour [U-13C]glucose infusion leads to systemic redistribution of the labeled glucose**, notably resulting in significant labeling of blood lactate due to peripheral metabolism (e.g., in muscle and liver). This means that astrocytes in their study were not only taking up labeled glucose but likely also labeled lactate, making it difficult to distinguish astrocytic glucose metabolism from systemic metabolic contributions. Furthermore, **astrocytes were isolated two hours after infusion**, providing ample time for metabolic turnover. The shown decreased fractional label in the GLUT1-deficient astrocytes can either be a result of decreased glucose breakdown (the author's interpretation) or of an increased uptake of unlabeled glucose, which would be seen in a case of increased metabolic activity in astrocytes. Given our observation that GLUT1-deficient astrocytes exhibit **higher glucose uptake and consumption**, it is highly plausible that there is an elevated uptake of unlabeled glucose and that labeled glucose-derived intermediates were metabolized more rapidly (to CO₂) or even lost during the isolation process.

Ardanaz et al. themselves noted a contradiction in their findings. While their PET data showed an **increase in whole-brain glucose metabolism**, their [U-13C]glucose labeling experiment suggested reduced astrocytic glucose flux. To reconcile this, they speculated that **neurons may be responsible** for the increased glucose metabolism, citing an increase in c-Fos activity in GLUT1ΔGFAP mice as potential evidence for enhanced neuronal activity. However, they do not provide direct evidence that neurons are taking up more glucose. Given that c-Fos activity does not necessarily correlate with neuronal glucose uptake, and that no neuronal glucose transporters were reported to be upregulated, this interpretation remains speculative. Additionally, their discussion briefly considers the possibility that microglia and oligodendrocytes could contribute to the PET signal, further highlighting the uncertainty surrounding the cellular source of increased glucose metabolism in their study.

In contrast, **our data provide direct evidence that astrocytes contribute significantly to the increased glucose metabolism**. Our biosensor imaging approach captures real-time glucose uptake and metabolism in astrocytes, avoiding the confounds associated with prolonged metabolic labeling and post-isolation analysis. The simplest and most consistent explanation is that GLUT1-deficient astrocytes increase their glucose uptake and utilization, which explains the elevated PET signal

observed Ardanaz et al.'s study. Thus, the discrepancy between our findings and theirs is best explained by **methodological differences in metabolic flux assessment**, rather than a fundamental difference in astrocytic glucose metabolism itself.

This detailed discussion is provided here for the reviewer's benefit, but in our manuscript, we will address this discrepancy succinctly by stating the following in the discussion of the manuscript:

Notably, our findings of enhanced glucose metabolism in astrocytes align well with a recent report that was published during the revision of our study, which demonstrated that astrocytic GLUT1 deletion is associated with increased whole-brain glucose metabolism, as identified by ^{18}F -FDG PET imaging⁶⁴. However, in contrast to both their PET results and our own findings, their analysis of $[\text{U-}^{13}\text{C}]$ glucose incorporation in isolated astrocytes suggested a reduction in astrocytic glucose metabolism⁶⁴. This discrepancy likely stems from methodological limitations. Their approach relied on prolonged systemic tracer infusion and a lengthy subsequent astrocyte isolation protocol, during which cellular metabolism remains active. This could lead to label dilution and metabolite turnover, potentially underestimating intracellular glucose metabolism. By contrast, our biosensor imaging provides direct, cell-specific evidence of increased glucose uptake and metabolism in GLUT1-deficient astrocytes. Hence, both our data and their PET findings⁶⁴ support the conclusion that astrocytes adaptively enhance their metabolic activity following GLUT1 loss.

Reviewer #3 (Remarks to the Author):

In this manuscript the authors investigate the role of astrocytic glucose transporter GLUT1 in astrocytic glucose metabolism as well as its contribution to brain homeostasis and responses to perturbation.

Using a combination of genetic, viral, and pharmacological manipulations both in slice and in vivo, the authors show that astrocyte-specific GLUT1 is dispensable to normal brain function (Figure 3) and behavior (Figure 2). Astrocytic deletion of GLUT1, the primary glucose transporter of this cell type, leads to no significant difference in basal glucose levels or in glucose uptake kinetics (Figure 3). However, there was a surprisingly significant increase in intracellular glucose levels in GLUT1-deleted astrocytes without compensation from other glucose transporters or connexin hemichannels (Figures 1, 4, S1, S2, and S6). The authors also posit that increased glucose metabolism in GLUT1-deleted astrocytes may be neuroprotective, as GLUT1 conditional knockout mice exhibit smaller infarct volumes following a model of stroke (Figure 5).

There are a few minor points that may be worth addressing:

- TRAP RNAseq. The authors write that there are only moderate differences in gene regulation between the genotypes as well as no significant changes related to glucose or monocarboxylate transport (Figure S2). The data presented in this figure – especially in the heatmap showing the top 13 differentially (spelled as “deferentially” in the manuscript) regulated genes are not strong support for these claims. I would recommend addition of genes related to glucose and monocarboxylate transport to the heatmap, with a note on the exact number of differentially regulated genes in either the main or figure text to quantify what the authors consider “moderate” gene differences. Alternatively, both up- and down-regulated DEGs could be plotted on a volcano plot (showing log₂FC and significance values) to show numbers of both significant and nonsignificantly changed genes.

Response: We appreciate the reviewer's suggestion and recognize that the phrasing in our results section and figure legend may have led to some misunderstanding. We have now revised both to provide a clearer and more precise description of our findings.

To clarify, Supplementary Fig. S2 already displays all significantly up- and downregulated differentially expressed genes (DEGs) based on an FDR < 0.05, ranked by their log₂ fold change between genotypes. We have now revised the figure legend to make this explicit and avoid any confusion. Regarding the suggestion to include genes related to glucose and monocarboxylate transport, we believe this would not enhance the heatmap's intended focus, as these genes are already specifically analyzed in Figure 11, where their expression levels are directly compared between genotypes. Including them in Figure S2b would therefore be redundant.

We also considered the suggestion of adding a volcano plot but feel that it would not provide additional information beyond what is already presented in Figure S2b, given that all significantly regulated genes are already displayed in the heatmap. Instead, we have now explicitly stated the total number of differentially expressed genes in the figure legend and main text, providing a clearer definition of what we consider "moderate" gene regulation differences.

Additionally, we have corrected the error to "differentially" in the manuscript, thanks for spotting this.

Here is the corrected results part in the manuscript:

Differential gene expression analysis identified 13 DEGs that were significantly (FDR < 0.05) up- or downregulated between the genotypes (Supplementary Fig. 2b), reflecting overall moderate changes in the translational profile. Gene Ontology (GO) enrichment analysis highlighted biological processes such as "vacuolar acidification," "positive regulation of glial cell differentiation," and "RNA processing" (Supplementary Fig. 2c). Notably, no overt changes were observed in processes related to energy metabolism, suggesting that GLUT1 deletion does not strongly impact the translation of metabolic genes in astrocytes.

- The concluding remarks on the manuscript's behavioral data. The authors suggest that "neurotransmission and synaptic plasticity likely remain intact despite loss of astrocytic GLUT1." While not a definite statement, it is still a bold one. Perhaps the authors could also query genes related to astrocyte synaptogenesis/long-term plasticity in the TRAP RNAseq? Alternatively, this comment could be removed or tempered – even without the speculation on neurotransmission and synaptic plasticity, the behavior results remain interesting.

Response: We thank the reviewer for pointing this out. We agree that our original statement on neurotransmission and synaptic plasticity was somewhat strong given that we did not directly assess synaptic function. To better reflect the scope of our findings, we have now revised this statement in the manuscript to a more tempered form, avoiding direct speculation on neurotransmission while still emphasizing the preserved behavioral outcomes.

Revised sentence:

"The absence of detectable impairments in learning and memory in GLUT1 cKO animals suggests that astrocytic GLUT1 deletion does not overtly disrupt cognitive function under these conditions."

- The use of Slc1a3 as an astrocyte-specific driver. Like most astrocyte-enriched targeting, Slc1a3 also is expressed in neural progenitors that give rise to neurons in the olfactory bulb and

hippocampus, and only targets a small fraction of all astrocytes (discussed by the authors in this manuscript's Discussion and reviewed nicely by PMID: 32042146). Have the authors taken advantage of this potential sparse labeling and compared recombined and nonrecombined astrocytes within the same animal?

Response: We thank the reviewer for this insightful comment. We initially considered comparing recombined and non-recombined astrocytes within the same animal, as we were interested in understanding how non-recombined astrocytes might compensate within the network. However, we quickly realized that this approach was not feasible due to the inherent limitations of our labeling strategy. Specifically, distinguishing non-recombined astrocytes from those that simply failed to express the tdTomato reporter due to incomplete AAV infection would introduce uncertainty. Since a lack of reporter expression could result from inefficient infection rather than the absence of recombination, this would make any direct comparison between recombined and non-recombined astrocytes unreliable.

To avoid potential misinterpretations, we opted not to perform this within-animal comparison. A more rigorous approach in future studies could involve crossing with a transgenic reporter line and inducing sparse recombination to reliably differentiate recombined from non-recombined astrocytes. However, generating and characterizing such a model, as well as replicating our metabolic imaging experiments in this context, would be beyond the scope of this revision.

- Gap junctions. The authors note in the discussion that inhibition of gap junctions and hemichannels led to a decrease in glucose levels, and that connexin-mediated regulation of astrocyte glucose metabolism remains unclear. Connexin-dependent astrocytic glucose transport across long distances has previously been reported in a mouse model of glaucoma (PMID: 32690710). Could it be that basal glucose levels are comparable in the GLUT1 cKO mice by compensation from nonrecombined astrocytes? It does seem like the distribution of values for individual cells (for example, in Figure 3e) is larger, perhaps reflecting two populations of recombined and nonrecombined cells? Additionally, the stroke lesion reduction experiment is interesting – are there intrinsic differences between how recombined and nonrecombined astrocytes work with or respond to glucose post-stroke?

Response: We thank the reviewer for this thoughtful comment and for pointing out the study by Cooper et al. (2020), which provides intriguing evidence for glucose shuttling through the astrocytic network in a mouse model of glaucoma. This study was one of the reasons we were particularly interested in investigating the role of gap junctions in glucose handling in GLUT1 cKO astrocytes.

We initially considered that GLUT1-deficient astrocytes might receive glucose from neighboring, non-recombined (~30%) astrocytes via gap junction coupling. To test this, we inhibited gap junctions and hemichannels using CBX and found that astrocytic glucose levels declined in both cKO and control slices. However, the interpretation of this finding is not straightforward. The drop in glucose levels with CBX does not necessarily indicate that gap junctions compensate for GLUT1 loss, as glucose uptake remained significantly enhanced in cKO astrocytes even in the presence of CBX when extracellular glucose was increased to 25 mM.

Previous studies have suggested that blocking gap junction coupling in cultured astrocytes can trigger increased glucose uptake via GLUT1 upregulation, facilitating astrocyte proliferation (Taberero et al., 2001, 2006; Sánchez-Alvarez et al., 2004). However, given that GLUT1 is absent in our cKO astrocytes, this suggests that connexin-mediated regulation of astrocytic glucose metabolism in adult astrocytes likely operates through mechanisms independent of GLUT1, an area requiring further investigation.

Regarding the suggestion that basal glucose levels in GLUT1 cKO mice might be maintained by compensation from non-recombined astrocytes, this remains an intriguing possibility. However, as discussed in our response above, we cannot reliably distinguish non-recombined from non-reporter-expressing astrocytes within the same animal. Importantly, in Fig. 3e, the analysis was performed specifically on tdTomato and glucose sensor co-expressing cells, meaning that the dataset should contain only recombined astrocytes. The observed variability in glucose levels across individual cells may reflect inherent heterogeneity in metabolic states rather than distinct populations of recombined versus non-recombined cells. Notably, as requested by the editorial team, the data in Fig. 3e are now presented in a box-and-whisker format.

The stroke lesion reduction finding remains particularly interesting, as it raises further questions about how the astrocyte network operates and contributes to glucose distribution under stroke conditions. We are particularly interested in understanding how gap junction coupling influences glucose dynamics post-stroke and whether recombined and non-recombined astrocytes differ in their metabolic response. As a next step, we plan to further investigate this question by including our inducible Cx30/Cx43 knockout model to specifically decouple the astrocytic network in a controlled manner. While these experiments are beyond the scope of the current study, we are excited to work on this in the near future.

Ultimately, these experiments support the idea that astrocytes are highly metabolically plastic in their ability to support proper neural function. The presented work in this manuscript has the potential to be interesting to groups studying the nervous system in both health and disease, and spans multiple levels of analysis.

Response: We sincerely thank the reviewer for their thoughtful feedback and for recognizing the broader impact of our work. We appreciate their insightful comments throughout the review process, which have helped strengthen the manuscript.

Reviewer #4 (Remarks to the Author):

The primary objective of the study by Thieren and colleagues was to determine the functional role of the glucose transporter Glut1 expressed by astrocytes. The authors present molecular and functional analyses performed in conditional mutant mice for astrocytic Glut1 and report intriguing findings including the preservation of sensorimotor and cognitive performances, an increase in glucose consumption by astrocytes as well as neuroprotection during an ischemic episode.

These data are unexpected as the 45 kd glut1 is thought to be the main (if not unique) glucose transporter expressed by astrocytes. The question of the membrane transporter through which glucose enters the cells of these KO mice remains enigmatic since the authors have shown that there is no compensation for the expression of other known transporters.

Overall, I consider that the **study was carried out rigorously by the authors and that, in particular, the validation of the cell specific glut1-deficient mouse model is convincing.**

I do, however, have a number of points and questions which, if they can be addressed by the authors, could increase the overall impact of their study.

Response: We thank the reviewer for the positive feedback.

1. In vivo evidence that glucose consumption is increased. The study of the mechanisms regulating

cerebral glucose metabolism is particularly relevant in vivo because it ensures that all the steps from blood glucose supply to local distribution and use are taken into account, including all the molecular and cellular players involved. It therefore seems to me that the current study would be greatly improved by the in vivo measurement of glucose consumption by glut1 KO mice. It would also be informative to measure the blood glucose levels of these animals. Such data would be very complementary to the one already obtained using FRET sensors in slices.

Response: We appreciate the reviewer's suggestion to provide in vivo evidence of increased glucose consumption in astrocytes. We fully agree that in vivo measurements of glucose metabolism are valuable, but obtaining astrocyte-specific glucose consumption data in vivo remains technically challenging. Current approaches, such as [¹³C]glucose tracer studies or PET imaging, lack cellular resolution and would not allow us to distinguish astrocytic from neuronal glucose uptake with the precision achieved in our slice-based approach using astrocyte-targeted glucose sensors.

Notably, a recent study by Ardanaz et al. (2024) using ¹⁸F-FDG PET imaging in a similar GLUT1 astrocyte knockout model demonstrated increased whole-brain glucose metabolism (see also our response to Reviewer #2). This finding aligns perfectly with our results, where FRET biosensor imaging in acute slices provides direct, high-resolution evidence of increased glucose consumption specifically in astrocytes. Given that PET imaging does not differentiate between cell types, our data offer critical cellular specificity, demonstrating that the increased glucose metabolism observed at the whole-brain level likely originates from astrocytes.

Regarding blood glucose levels, we have now included measurements we had initially collected in the revised manuscript (**NEW Supplementary Fig. S3j**). We did not observe significant differences in blood glucose levels between GLUT1 cKO and control mice, indicating that the metabolic adaptations in GLUT1-deficient astrocytes occur without systemic alterations in glucose homeostasis.

Overall, we are very keen on establishing transport-stop assays to directly measure glucose flux in vivo; however, this requires further technical optimization and is not feasible within the scope of this revision. We fully acknowledge the value of additional in vivo approaches, but we hope the reviewer appreciates that our current data provide compelling, astrocyte-specific evidence that GLUT1 deletion enhances glucose uptake and metabolism specifically in astrocytes.

2. Cellular expression of the protein Glut1. The authors could consider using immunogold electron microscopy to convincingly demonstrate the loss of Glut1 expression in astrocyte membranes and the potential relocation of residual transporters.

Response: We appreciate the reviewer's suggestion to use immunogold electron microscopy to assess GLUT1 expression at the ultrastructural level. However, we are confident that our current approach provides compelling evidence that GLUT1 is effectively eliminated from astrocytes following recombination.

We performed our analyses 60 days after tamoxifen administration, a timeframe that should allow for complete degradation of any residual GLUT1 protein. For comparison, the estimated half-life of GLUT1 in endothelial cells is approximately one week or less (Jais et al., 2016; Veys et al., 2020). Studies on inducible endothelial GLUT1 deletion result in severe neuroinflammation, neuronal loss, and lethality within just one week of tamoxifen treatment (Jais et al., 2016; Veys et al., 2020). If GLUT1 persisted for prolonged periods post-recombination, such rapid pathology would not be observed, suggesting that GLUT1 protein turnover is highly efficient following gene deletion.

While we recognize that GLUT1 turnover may differ across cell types, we have directly validated its loss in astrocytes using both Western blotting and immunohistochemistry, which demonstrate clear depletion of GLUT1 protein. These complementary techniques provide both quantitative and spatial confirmation of successful GLUT1 ablation. While immunogold electron microscopy can provide ultrastructural insights, it is not an optimal method for robust quantification of membrane protein expression, given its low labeling efficiency and inherent technical variability. For this reason, we believe that our Western blot and immunohistochemistry data provide the most reliable assessment of GLUT1 loss.

3. Glycogen content. Glycogen is stored in astrocytes and is thought to provide lactate as an energy source to neurons through monocarboxylate transporters (MCTs) to maintain memory formation. It would be of particular interest to determine whether glycogen levels are altered in glut1 KO mice.

Response: We appreciate the reviewer's comment regarding glycogen content and its role in memory formation. Previous studies, including Suzuki et al. (2011) and our own work (Zuend et al., 2020), have demonstrated that glycogen-derived lactate plays a role in learning and memory via MCTs. However, in our study, we did not observe any deficits in learning or memory in Barnes maze or passive avoidance tests, indicating that astrocyte-derived metabolic support is functionally preserved despite GLUT1 deletion.

Moreover, our stroke experiments suggest that astrocytic metabolic support may not only be maintained but actually enhanced. The significant neuroprotection observed in GLUT1 cKO mice following ischemia suggests that astrocytes are still able to provide metabolic support under stress conditions. If glycogen-derived lactate metabolism were impaired, we might have expected functional deficits or increased neuronal vulnerability, neither of which were observed.

In line with these findings, our RNA-seq analysis did not reveal any significant changes in genes related to glycogen synthesis or breakdown, further supporting the notion that glycogen metabolism likely remains unaffected in GLUT1 cKO astrocytes. We believe that investigating glycogen content would be unlikely to provide additional mechanistic insight, as learning and memory function remain unaffected in GLUT1 cKO mice.

Minor points.

1. The sentence starting line 67 "The structure and function...also due to its implication" should be reworded.

Response: We fully agree with the reviewer. This sentence/section has been reworded.

2. Figure 4. The rates of glucose rise (Fig. 4b) as well as the glucose changes (Fig. 4c) vary greatly between cells. Did the authors use the number of cells (n= 88 and 55) for their statistics or the number of experiments?

Response: Indeed, statistical analysis was performed using the number of cells (n = 88 and 53), as stated in the figure legend. While data were derived from multiple independent experiments across different animals to account for biological variability, we acknowledge that analyzing data at the experiment/animal level provides a more robust approach. To address this, we have reanalyzed the data using a linear mixed model (LMM) to account for the nested data structure, considering cells

within distinct brain slices and animals. The p-value, obtained from the nested LMM analysis, is depicted in the figure and explicitly stated in the figure legend.

Furthermore, to comply with the editorial requests the data are now presented in a box-and-whisker format.

3. Suppl. Fig.3. I do not quite understand how semi-quantification was achieved using GFAP, S100beta, IBA1 and NeuN immunostaining using a very low number of animals.

Response: We appreciate the reviewer's concern. We acknowledge that the variability in GFAP and IBA1 data was initially high, making quantification with $n = 3$ animals per genotype less ideal. To address this, we have increased the sample size to $n = 7$ vs 6 animals. While some variability remains, we still observe no differences between genotypes. For S100 β and NeuN analysis, we originally used $n = 4$ animals per genotype. As these markers showed lower variability, we believe this sample size is sufficient to confirm the absence of differences, given that no trends indicating genotype-specific effects were observed. Furthermore, for all stainings, we analyzed at least three brain sections per animal, with three images per section, ensuring a robust and representative quantification approach.

Thieren et al., NCOMMS-24-42691A Point-to-point response

The reviewers' comments are cited directly from the decision letter. Below are the individual comments (black) and our responses (blue).

Reviewer #1 (Remarks to the Author):

Authors have addressed all my comments satisfactorily and I don't have any further remarks. I look forward to seeing this beautiful study published.

Response: We thank the reviewer for the kind words and appreciation of our work.

Reviewer #2 (Remarks to the Author):

In my view the authors adequately addressed the comments raised by the 4 reviewers.

Response: We thank the reviewer for the positive assessment and support.

Reviewer #3 (Remarks to the Author):

In this manuscript the authors investigate the role of astrocytic glucose transporter GLUT1 in astrocytic glucose metabolism as well as its contribution to brain homeostasis and responses to perturbation. Using a combination of genetic, viral, and pharmacological manipulations both in slice and in vivo, the authors show that astrocyte-specific GLUT1 is dispensable to normal brain function (Figure 3) and behavior (Figure 2). Astrocytic deletion of GLUT1, the primary glucose transporter of this cell type, leads to no significant difference in basal glucose levels or in glucose uptake kinetics (Figure 3). However, there was a surprisingly significant increase in intracellular glucose levels in GLUT1-deleted astrocytes without compensation from other glucose transporters or connexin hemichannels (Figures 1, 4, S1, S2, and S6). The authors also posit that increased glucose metabolism in GLUT1-deleted astrocytes may be neuroprotective, as GLUT1 conditional knockout mice exhibit smaller infarct volumes following a model of stroke (Figure 5). Ultimately, these experiments support the idea that astrocytes are highly metabolically plastic in their ability to support proper neural function. The presented work in this manuscript has the potential to be interesting to groups studying the nervous system in both health and disease, and spans multiple levels of analysis.

The authors have addressed my concerns through this revision sufficiently. I have no further comments.

Response: We thank the reviewer for recognizing the revisions made and appreciate the encouraging feedback.